



# Constraining stochastic 3-D structural geological models with topology information using Approximate Bayesian Computation using GemPy 2.1

Alexander Schaaf[1,2], Miguel de la Varga[2], Florian Wellmann[2], and Clare E. Bond[1]

[1]Geology and Petroleum Geology, School of Geosciences, University of Aberdeen, AB24 3UE, UK
[2]Computational Geoscience and Reservoir Engineering, RWTH Aachen University, Aachen, Germany

**Correspondence:** a.schaaf@abdn.ac.uk

**Abstract.** Structural geomodeling is a key technology for the visualization and quantification of subsurface systems. Given the limited data and the resulting necessity for geological interpretation to construct these geomodels, uncertainty is pervasive and traditionally unquantified. Probabilistic geomodeling allows for the simulation of uncertainties by automatically constructing geomodels from perturbed input data sampled from probability distributions. But random sampling of input parameters can

lead to construction of geomodels that are unrealistic, either due to modeling artefacts or by not matching known information about the regional geology of the modeled system. We present here a method to incorporate geological information in the form of geomodel topology into stochastic simulations to constrain resulting probabilistic geomodel ensembles. Simulated geomodel realisations are checked against topology information using a likelihood-free Approximate Bayesian Computation approach. We demonstrate how we can learn our input data parameter (prior) distributions on topology information in two

experiments: (1) A synthetic geomodel using a rejection sampling scheme (ABC-REJ) to demonstrate the approach; (2) A geomodel of a subset of the Gullfaks field in the North Sea, comparing both rejection sampling and a Sequential Monte Carlo sampler (ABC-SMC). We also discuss possible speed-ups of using more advanced sampling techniques to avoid simulation of unfeasible geomodels in the first place. Results demonstrate the feasibility to use topology as a summary statistic, to restrict the generation of model ensembles with additional geological information and to obtain improved ensembles of probable

geomodels using stochastic simulation methods.

## 1   Introduction

Structural geomodeling is an elemental part of visualizing and quantifying geological systems (Wellmann and Caumon, 2018). Topology relationships in geological systems (e.g. how layers are connected to each other stratigraphically, or their across-

fault connectivity) are important constraints for fundamental geological processes, such as fluid or heat flow (Thiele et al., 2016a, b). Each unique interpretation (model) of a geological setting has a specific topology graph. And as geology is not only





an experimental, but also an interpretive and historical science (Frodeman, 1995), the deduction of the geomodel - often from sparse amounts of data – can inherently lead to numerous valid geological interpretations (Bond et al., 2007), which themselves can lead to equally numerous topology graphs. This aspect is compounded by the complex nature of geological systems and

interpretation bias by geoscientists in the explicit creation of geomodels (Bond et al., 2007; Polson and Curtis, 2010; Bond, 2015). It also leads to the creation, and favouring, of specific models that fit expectations and prior knowledge (Baddeley et al., 2004), rather than consideration of the full range of possible models. However, methodologies to create models often focus on the creation of a single deterministic model (Bond et al., 2008) and a lack of systematically considering data uncertainty (Thore et al., 2002; Tacher et al., 2006; Bardossy and Fodor, 2013) . These facts call for the development of alternative approaches.

The increasing development of implicit modeling algorithms (Mallet, 2004; Hillier et al., 2014; Laurent et al., 2016) allows for the creation of vast structural geomodel ensembles by making use of interpolation functions, which makes the analysis and visualization of uncertainty using probabilistic simulation approaches possible (Bistacchi et al., 2008; Suzuki et al., 2008; Wellmann et al., 2010; Lindsay et al., 2012; Wellmann and Regenauer-Lieb, 2012; Wellmann, 2013).

But the mathematical nature of implicit modeling, in combination with the use of a probabilistic modeling process, often

leads to geologically unsound model realizations and modeling artifacts. Additionally, the modeling algorithms only take a limited set of input data types, e.g. layer interface locations and structural orientation data, which significantly limit the amount of geological information that can be included in the modeling process. de la Varga and Wellmann (2016) and Wellmann et al. (2017) showed how Bayesian inference can be used to reduce uncertainty and modeling artifacts in both synthetic and real, implicit, structural geomodel ensembles. Their concept uses supplemental geological information (e.g. layer thicknesses or

fault offsets) in the form of *likelihood functions* to constrain stochastic geomodel ensembles. But the question of how to acquire suitable likelihood functions for specific geological systems and diverse types of prior geological knowledge and reasoning remains. Likelihood functions essentially represent information in a probabilistic mathematical form. This information can be available numerical data, such as information about the range of possible layer thicknesses in a depositional setting.

But geological expert knowledge contains much more information that is vital to model creation, such as understanding the

geological processes that result in the thickening and thinning of sedimentary deposits and their relative spatial distribution. One key knowledge-based input into geomodeling is the understanding of the kinematic evolution of the rock units into their present configuration. While kinematic modelling software exists (see Groshong et al., 2012; Brandes and Tanner, 2014, for reviews), it is limited to 'end-member' kinematic models' resulting in geometrical deformations defined by few parameters, and not taking into account a range of other factors, not least the mechanics of the different units (Butler et al., 2018). But

we can capture certain kinematics using topology information—for example the across-fault connectivity of layers, where extensional deformation leads to fundamentally different topological relationships than does compressional deformation (see Fig. 1).

We therefore hypothesize that topological information about a geological system can be used as a meaningful constraint for probabilistic 3-D geomodeling outputs.

But this topological information is difficult to incorporate into the mathematical foundations of implicit modeling functions and is highly case-dependant. As the origin of topological information is generally qualitative, obtaining a suitable likelihood





function that can be used in a Bayesian inference is considered intractable, apart maybe from time- and cost-consuming expert elicitation (Curtis and Wood, 2004). This work tries instead to approximate the (Bayesian) posterior geomodel ensemble that incorporates both the geological input data and the topology information using an Approximate Bayesian Computation (ABC)
approach for a likelihood-free approximation of the posterior.

To test this approach we designed two distinct experiments, one synthetic and one case study:

1. We construct a synthetic fault model and explore its topological uncertainty. We do this by describing our input data not as fixed parameters, but as probability distributions. We then use Monte Carlo sampling to obtain input data from which geomodels are constructed. We then show how a single topology graph can be used as a summary statistic in an
ABC-rejection scheme to approximate the posterior model ensemble that honours the added information.

2. To test the same ABC approach on a real-world dataset, we apply it to a model extracted from a seismic interpretation of the North Sea Gullfaks field. We also explore a more advanced sampling technique to demonstrate possibilities for reducing the computational costs of the method

In the following section we will give an overview of the applied implicit geomodeling approach, the basic concept of
70 Bayesian inference and its use in probabilistic geomodeling, as well as the idea behind Approximate Bayesian Computation. We further describe how we analyze model topology and use it as a summary statistic. We will then introduce, in detail, both the synthetic fault model and the case study, followed by a comprehensive discussion of our findings.

## 2 Materials and Methods

### 2.1 Implicit Geomodeling

Several approaches exist for creating structural geomodels, which can be separated into three main categories: (a) interpolation, (b) kinematic methods and (c) process simulation. The interpolation of surfaces and volumes from spatial data is currently the most widely used approach in geosciences, especially manually, which requires robust knowledge of the geological setting and extensive amounts of data in order to robustly approximate reality. Additionally, highly complex structures such as extensive fault networks and repeatedly folded areas are challenging to recreate using current interpolation methods (Jessell et al., 2014;
Wellmann et al., 2016; Laurent et al., 2016).

The open-source, Python-based implicit modeling package GemPy[1] (de la Varga et al., 2019) is used here. It is based on the work of Lajaunie et al. (1997) and Calcagno et al. (2008), and allows the interpolation of geological interface position and plane orientation data by using a scalar field method in combination with cokriging (Chilès et al., 2004). For a detailed overview of the algorithm and the functionality of GemPy, we refer the reader to de la Varga et al. (2019).

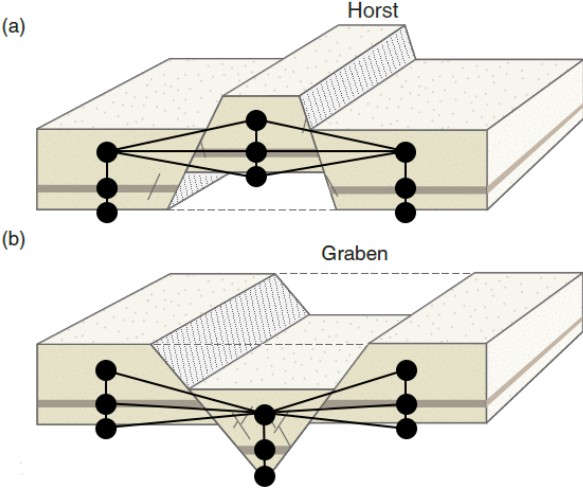

**Figure 1.** Idealized Horst (a) and Graben (b) structures with topology graph overlay, showing the difference in graph structure for different tectonic settings (modified from Fossen, 2010). The black nodes represent the centroids of the geobodies and the black edges the topology connections, together building a topology graph.

## 2.2 Geological Topology

Topology, referring to "properties of space that are maintained under continuous deformation, such as adjacency, overlap or separation" (Thiele et al., 2016a; Crossley, 2006), is a highly relevant concept in structural geology, as it provides a useful description of the relations between stratigraphic units across layer interfaces, faults or the contact to an intrusive body. Generally, eight binary topological relationships can exist between three-dimensional objects (Egenhofer, 1990), while a total of 69 relations are possible between *simple* lines, surfaces and bodies (e.g. surfaces without holes; see Zlatanova, 2000). From these eight Egenhofer-Herring relationships, *meets* (i.e. adjacency) is the most relevant one for describing structural and stratigraphic relationships, such as across-fault connectivity of layers (see Fig. 1). The topology relationships of geological models can be represented by an adjacency graph, which represents topological units as individual nodes and their connections by edges (see Fig. 1). The adjacency topology of geological structures is highly dependent on deformation: compressional deformation leads to different connectivities in the topology graph than does extensional, but even within the same type of deformation they can lead to different topologies—as visualized by the Horst and Graben structures in Figure 1. Not only does the type of deformation have an important influence on the systems topology, but also the quantity—e.g. the fault throw. For an in-depth introduction and discussion of topology in geology see Thiele et al. (2016a) for the fundamental theory and Thiele et al. (2016b) and also Pakyuz-Charrier et al. (2019) for the influence of structural uncertainty on geomodel topology.

[1]URL: github.com/cgre-aachen/gempy



### 2.2.1 Computing geomodel topology

To compute the geomodel topology with the necessary computational efficiency to conduct a feasible stochastic simulation of realistic geomodels, we implemented a topology algorithm using `theano` (Theano Development Team et al., 2016) into the core of `GemPy`. This enables the topology computation to run alongside the geomodel interpolation on graphical processing units (GPUs). As `theano` is a highly optimized linear algebra library, the employed method is mainly focused on utilizing matrix operations for the computation of the geomodel topology. When the implicit geomodel is discretized using a regular grid, it becomes a 3-D matrix of lithology IDs $L$ (Fig. 2a), which we use for the calculation of the geomodel topology. For each geomodel we also have access to the 3-D boolean matrices $F_n$ for each fault, representing the two sides of the respective fault by two ascending consecutive integers (Fig. 2b). Given these two input data, we compute the geomodel topology as follows:

1. The lithology matrix $L$ and the summed fault matrices $\sum_{i=1}^{n_{\text{fault}}} F_i$, where $n_{\text{fault}}$ is the total number of faults in the geomodel, are combined into a matrix where each lithology in each fault block is represented by its own unique integer, referred to as the topology labels matrix $T$ (see Fig. 2c):

$$T = L + n_{\text{lith}} \sum_{i=1}^{n_{\text{fault}}} F_i \tag{1}$$

with $n_{\text{lith}}$ being the total number of lithology IDs in the geomodel.

2. The topology labels matrix $T$ is then shifted twice (forward and backward) along each axis X, Y and Z. The two resulting shifted matrices $S_1$ and $S_2$ along each axis are then subtracted from each other to result in a difference matrix $D$, in which only the cells along a lithology or fault boundary are non-zero (Fig. 3).

3. The topology labels matrix $T$ is then evaluated at all non-zero cells of $D$ to obtain the two topology labels $n_a, n_b$ of each topological connection (reffered to as an edge $e$) in the geobody, which are stored in a set of unique edges $E$ representing the geomodels topology. For the example shown in Figure 2 and 3 the abbreviated set is $E = \{(0,4), (0,5), (0,1), ..., (3,7)\}$.

This method of topology calculation works on regular grids, which imposes a strong bias on the result: if the main lithological and structural features are not aligned with the grid orientation, the resulting topology graph could thus contain (or miss) connections. For a more detailed discussion on the effects of model discretization see Wellmann and Caumon (2018).

## 2.3 Stochastic Modelling Approach

### 2.3.1 Bayesian Inference

Bayesian inference is fundamentally different to the classical frequentist approach of inference. It treats probabilities as *degrees of certainty* of a parameter $\theta$, which is inherently considered to be a random variable itself (Bolstad, 2009; VanderPlas, 2014). It is based on *Bayes' theorem* (Eq. 2), which allows to update a given probability - the *prior probability* $p(\theta)$ of a parameter $\theta$ - after the occurrence of a connected event (Bolstad, 2009). This updating process relies on the use of a *likelihood function*





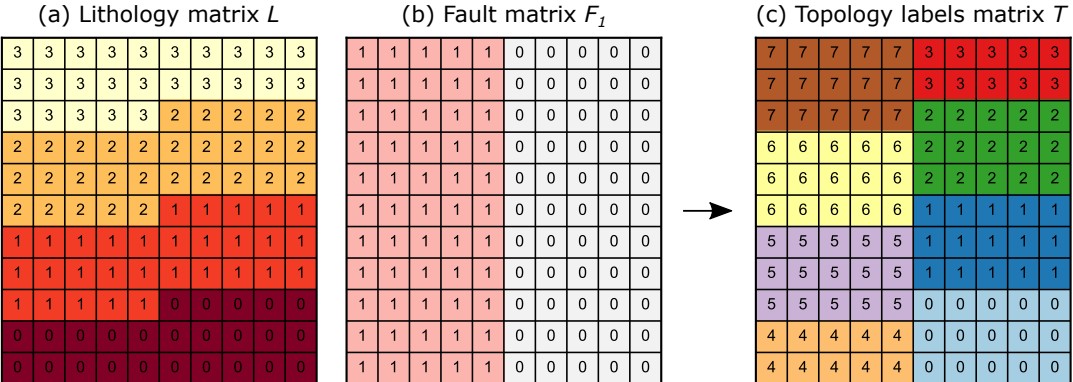

**Figure 2.** (a) Lithology matrix $L$ of an example 2D geomodel that consists of four layers and a vertical fault in the center; (b) Fault matrix $F$ of the geomodel; (c) Topology labels matrix $T$ of the geomodel.

**Figure 3.** Vertical (a) and horizontal (b) difference matrix $D$ showing all cells (red) in the shifted matrices $S_1$ and $S_2$ which are next to the interface between two different layers or of any layers across a fault. The highlighted (yellow) part shows the area in which the implicit interface must be located.

$p(y|\theta)$, representing the probability distribution of the observed data $y$ of the occurring event. It is used to condition the *prior* into the *posterior distribution* $p(\theta|y)$, which represents the degree of certainty over the parameter $\theta$ after the occurrence of the event and its observed data $y$.

$$p(\theta|y) = \frac{p(y|\theta)\,p(\theta)}{\int p(y|\theta)\,p(\theta)\,\mathrm{d}\theta} \qquad (2)$$

For the use in geomodeling, these parameters can be seen as (de la Varga and Wellmann, 2016; Gelman et al., 2013):

– **Model parameters** $\theta$: The model-defining parameters (e.g. layer interface positions, dip or fault parameters used for the interpolation of the geomodel), which can be either deterministic (thus be exactly defined and known) or probabilistic. The latter represent uncertain parameters, which is expressed in the form of probability distributions (e.g. a normal distribution expressing the uncertainty of the vertical subsurface position of a layer interface);





– **Observed data** $y$: Represents additional measurements or observations, which should enhance the model definition by providing additional information with the goal to reduce model uncertainty or enable the comparison of the model to reality (e.g. by comparing geophysical potential-field measurements with the according forward simulation on the basis of a geomodel). In this work we use topology information in the form of a topology adjacency graph as the "observed data";

– **Likelihood functions** $p(y|\theta)$: These form the relationship between the model parameters $\theta$ and the observed data $y$. Essentially, this function describes the likelihood for the parameters $\theta$ for a given observation $y$ (e.g. MacKay and Kay, 2003). In the case of structural modeling, this essentially means that we compute the geomodel from the input parameters $\theta$ and compare model predictions (e.g. the thickness of a certain layer at a certain position), with additional observed data. The likelihood of the parameter $\theta$ is then encoded in the likelihood function.

While constructing meaningful likelihood functions for physical properties such as layer thickness or geobody volume from observed data is straight forward (de la Varga and Wellmann, 2016), we have no proper framework to construct them for more abstract or "soft data", such as our understanding of the geological setting, or the topology relationships of our layers across faults or unconformities. For this reason, we chose to pursue a likelihood-free method to estimate our posterior distributions given abstract geological information: Approximate Bayesian Computation.

### 2.3.2 Approximate Bayesian Computation

Geoscientists often have extensive implicit knowledge of the geological settings (e.g. our understanding of the tectonics of a system), but only a limited amount of this knowledge can be incorporated into the geological interpolation function (Wellmann and Caumon, 2018). Additionally, it is often difficult to define formal likelihood functions for geological knowledge, as required for conventional Bayesian inference methods. A less formal but valid alternative approach is to approximate the posterior distributions using Approximate Bayesian Computation (ABC) methods. These methods are also referred to as likelihood-free inference methods (Marin et al., 2012), ABC methods evaluate the distance of stochastically generated models to our additional data using one or multiple summary statistics $S$ (e.g. model topology), instead of a probabilistic likelihood function.

To obtain the approximate posterior distribution we need to sample from our prior parameter distributions, plug the values into our simulator functions (our geomodeling software), compute the summary statistic $y$ (geomodel topology) and evaluate its distance to our observed summary statistic (data) $\hat{y}$ (e.g. a geomodel topology graph). The most fundamental sampling scheme for ABC is based on rejection sampling (ABC-REJ; see Algorithm 1), for which the distance between our simulated data $y$ and observed data $\hat{y}$ is calculated using a distance function of the summary statistics $d\big(S(\hat{y}), S(y(\theta'))\big)$. The simulated model is accepted if the distance is below a user-specified error bound $\epsilon \geq 0$ (Sadegh and Vrugt, 2014), or else rejected. The accepted samples form the approximate posterior. Thus, this method circumvents the need to specify a likelihood function for our additional data, while still approximating the posterior distributions incorporating the information of both our priors and our additional information (Sunnåker et al., 2013). Within this work we use the Jaccard index $(1 - J)$ as a distance function between topology graphs.



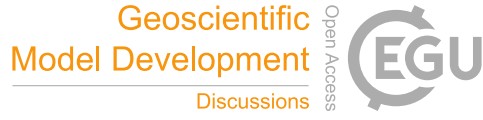

---

**Algorithm 1** ABC-REJ

---

**for** $i = 0$ to $N$ **do**

    **while** $d\big(S(\hat{y}), S(y(\theta')))\big) > \epsilon$ **do**

        Draw sample $\theta'$ from priors $p(\theta)$

        Simulate geomodel $y(\theta')$

        Compute geomodel topology $S(y(\theta'))$

        Calculate $d\big(S(\hat{y}), S(y(\theta')))\big)$

    **end while**

**end for**

---

A more advanced sampling scheme for ABC is Sequential Monte Carlo sampling (ABC-SMC). In its simplest form it can be seen as an extension of rejection sampling, by chaining rejection sampling simulations together (each referred to as an *epoch*). During the first epoch of rejection sampling, a large error threshold $\epsilon_1$ is used while sampling from the prior distributions $p(\theta)$. The accepted samples, forming the posterior distributions of the first epoch, form the updated priors of the second epoch by

replacing the priors with the kernel density estimation $\hat{f}_h(\theta_{accepted})$ of the posterior samples. Iteratively, with every epoch, the error threshold $\epsilon$ is reduced to the target value (e.g. $\epsilon = 0$) to obtain the final posterior sample. Thus, every epoch, the sampler 'learns' from the previous epoch by adjusting the prior distributions further towards the posterior distributions. As ABC-REJ tends to suffer from potentially low computational efficiency when using low error thresholds $\epsilon$, the iterative shrinking paired with adjustment of the prior distributions can potentially obtain the approximate posterior much more quickly. We apply this

sampling scheme to our Gullfaks case study to show the potential speed-ups.

---

**Algorithm 2** ABC-SMC

---

**for** $\epsilon$ in $\{\epsilon_1, \epsilon_2, ..., \epsilon_M\}$ **do**

    **for** $i = 0$ to $N$ **do**

        **while** $d\big(S(\hat{y}), S(y(\theta')))\big) > \epsilon$ **do**

            Draw sample $\theta'$ from priors $p(\theta)$

            Simulate geomodel $y(\theta')$

            Compute geomodel topology $S(y(\theta'))$

            Calculate $d\big(S(\hat{y}), S(y(\theta')))\big)$

        **end while**

    **end for**

    Replace priors $p(\theta)$ with KDE $\hat{f}_h(\theta_{accepted})$

**end for**

---





## 2.4 Topology distance functions

To use geomodel topology as a constraint for probabilistic geomodels in an ABC framework, we need a consistent way of comparing geomodel topologies—i.e. suitable distance functions. We consider here three possible comparison methods:

1. **Presence or abscence of defined connections:** As the relational topology information is captured in adjacency graphs, the most fundamental approach is to check if two relevant nodes $n_1$ and $n_2$ (e.g. representing two regions in the model) share an edge $e = (n_1, n_2)$ (are adjacent), and if this edge exists in both models. This is the most simple way of comparing specific aspects of relational topology between geomodels. This approach can be viewed as a boolean comparison: *True* if the given edge exists in both models, *False* if not. This also enables the direct comparison of $i$ multiple edges, which would result in a vector of $i$ boolean statements for each comparison $[e_1, e_2, \ldots, e_i]$.

2. **Comparing entire graphs:** To compare topology graphs as a whole, Thiele et al. (2016b) describe the use of the Jaccard index (Jaccard, 1912). It can be used to compare the similarity of sets by creating the ratio of the intersection and union of two graphs $A$ and $B$:

$$J(A, B) = \frac{|A \cap B|}{|A \cup B|} \tag{3}$$

For two topology graphs $A$ and $B$, this means we calculate the ratio of edges (representing connected regions) shared in both (intersection: $A \cap B$) and their total combined number of edges (union: $A \cup B$). This ratio can be used to efficiently identify all unique topology graphs in a given ensemble, as only an identical pair of graphs results in a Jaccard index of $J(A, B) = 1$. A comparison using the Jaccard index yields ratios of integers, thus a discrete comparison. This method also allows specifying a tolerance $0 < \epsilon < 1$ for model acceptance, i.e. to accept models within the range $1 - \epsilon \leq J \leq 1$.

3. **Contact area:** Comparing the number of actual edge pixels (or voxels), representing the area of the contact $A_e$ between two geobodies could yield a more granular comparison that allows to take into accounts trends of the contact size. Thus the ABC error tolerance $\epsilon$ could be used to reject geomodels where certain topological contact areas are above and/or below a certain value $A_e - \epsilon_{low} \leq A_e \leq A_e + \epsilon_{high}$.

## 2.5 Quantifying Uncertainty using Shannon Entropy

Stochastic simulations yield vast ensembles of geomodel realizations and their variability (and thus uncertainty) needs to be analyzed and understood. The uncertainty of a single geological entity (e.g. a layer or a fault) can be estimated from its frequency of occurrence in each single geomodel voxel. In order to analyze the whole geomodel uncertainty at once, more sophisticated measures can be applied: the concept of *Shannon entropy H* can be used in a spatial context to evaluate the uncertainty of an entire geomodel ensemble at once, as described by Wellmann and Regenauer-Lieb (2012). Their concept is based on concepts from information theory, derived by Shannon (1948), and further on the concept of fuzziness established





by Zadeh (1965) and De Luca and Termini (1972). If applied to a fuzzy set[2] $f \in [0, 1]$ in a grid, the measure should only be $0$ if every grid cell is either $0$ or $1$ everywhere (thus the grid having no uncertainty anywhere, meaning we are absolutely certain about the lithology at this position), and should have its maximum value when $f = 0.5$ for all grid cells (meaning all outcomes are equally likely, which represents the highest uncertainty possible: every lithology is equally likely to be present at this position). The resulting equation is:

$$
\begin{aligned}
H_m = -\frac{1}{N} \sum_{x=1}^{N} &\Big[ p_m(x) \log_2 \big( p_m(x) \big) \\
&+ \big( 1 - p_m(x) \big) \log_2 \big( 1 - p_m(x) \big) \Big]
\end{aligned}
\tag{4}
$$

where we denote the fuzzy set $f$ as the probability $p_m$ of an outcome $m \in M$ of a cell $x$, and $H_m$ being the Shannon entropy normalized by the total number of cells $N$. The average model entropy $\overline{H}$ can also be evaluated by:

$$
\overline{H} = -\frac{1}{N} \sum_{x=1}^{N} H(x)
\tag{5}
$$

Which makes the average model entropy equal to $0$ if all cells $x$ have only one possible outcome (no uncertainty), and reaching its maximum when all outcomes are equally likely for all cells of the model (maximum uncertainty).

## 2.6 Experiment Design

### 2.6.1 Synthetic Fault Model

As a proof of concept we show how ABC can be used to incorporate geological knowledge and reasoning into an uncertain synthetic geomodel. This model represents a folded layer cake stratigraphy that is cut by a N-S striking normal fault to represent an idealised reservoir scenario frequently encountered in the energy industry (see Fig. 4a).

The prior parametrization is schematically visualized in Figure 4b and consists of two different kinds of uncertain parameters: (i) vertical location of the layer and fault interfaces and (ii) lateral location of the fault interface, with the specific parametrization displayed in Table 1 in the Appendix. Two separate simulations were run for this experiment so we can see how topology can constrain an uncertain geomodel compared to the Monte Carlo simulation of uncertainties:

1. A Monte Carlo simulation of the prior parameters to evaluate the uncertainty in the resulting geomodel ensemble consisting of 2000 generated models. This represents our 'base case' uncertainty without any constraints.

2. An Approximate Bayesian Computation using the initial model topology graph (see Fig. 4c) to represent our geological knowledge. We are employing a rejection sampling scheme (ABC-REJ) with an error tolerance of $\epsilon = 0$ to obtain $500$ generated posterior models. Thus, the resulting posterior geomodel ensemble will contain only samples with matching topology graphs.



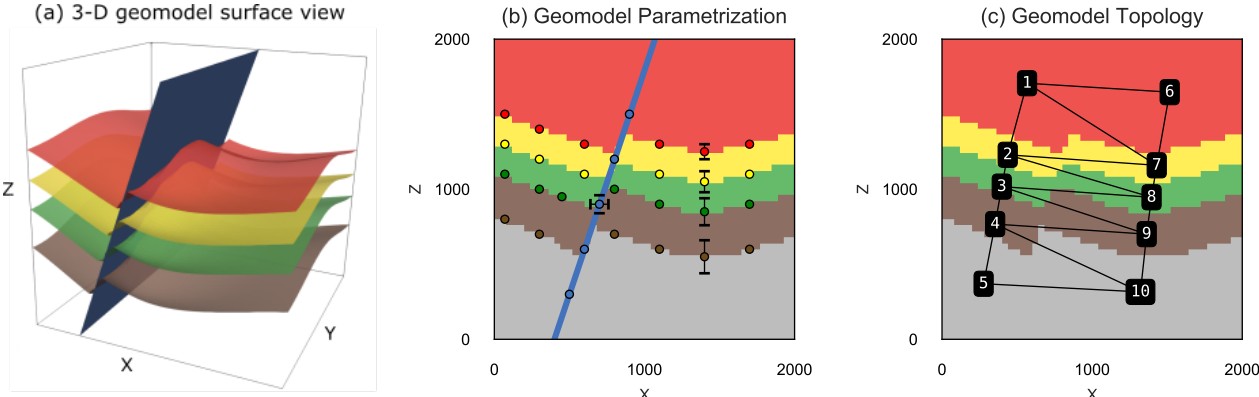

**Figure 4.** (a) 3D view of the synthetic fault model, with top surfaces of the four lithologies shown and the fault surface in blue; (b) XZ-slice through the center of the discretized model showing partial input data (for visual brevity) and example standard deviations of prior parameters used for the stochastic simulation; (c) Model overlaid with its topology graph used as our summary statistic for the ABC.

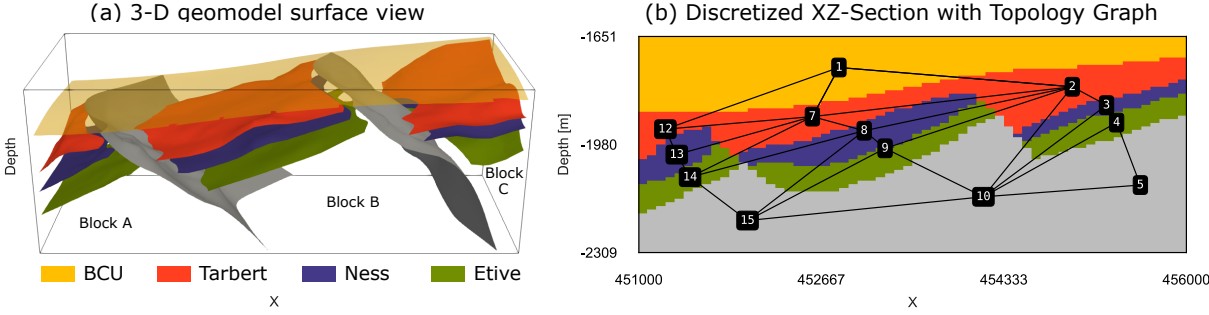

**Figure 5.** (a) 3D view of the Gullfaks geomodel used as mean prior model in our case study; (b) XZ-section through the discretized geomodel with overlaid topology graph showing the inter- and intra-fault block relations of geobodies.

### 2.6.2 Case Study: The Gullfaks Field

To demonstrate the applicability of the method to real datasets we apply it to a model of part of the Gullfaks Field, located in the northern North Sea. The field is located in the western part of the Viking Graben, and consists of the NNE-SSW-trending 10-25 km wide Gullfaks fault block (Fossen and Hesthammer, 1998). For a detailed overview of the regional and structural geology we refer to Fossen and Rørnes (1996); Fossen and Hesthammer (1998); Fossen et al. (2000); Schaaf and Bond (2019).

For the experiment, we constructed a base geomodel (Fig. 5a) founded in an interpretation of the training data set provided with the seismic interpretation software Petrel™. We have chosen a relatively simple subset of the interpretation, containing 2 faults, three horizon tops Tarbert (red), Ness (purple) and Etive (green), and the Base Cretaceous Unconformity (BCU, yellow).

²i.e. a non-binary set with real numbers in-between the two interval boundaries.





To create the geomodel, we exported the corresponding seismic interpretation data from Petrel and imported them into Python.
The surface interpretations were then decimated down to $510$ surface points and $187$ surface orientations, via a target reduction
of $80\,\%$ per fault block or surface using the VTK-based decimation functionality of `pyvista` (Sullivan and Kaszynski, 2019),
to retain the best possible surface shape while allowing fast implicit geomodel construction times in `GemPy`.

The prior parametrization consists of two different kinds of uncertain parameters: (i) vertical location of the layer interfaces
for within each fault block; (ii) the lateral location of the fault interfaces. This parametrization is similar to the synthetic
fault model (all specifications are listed in Table 2 in the Appendix). This parametrization was chosen due to its ease of
implementation and to demonstrate how simplified uncertainty modeling can lead to highly uncertain results, especially
regarding the topology graphs of the resulting geomodel ensembles in real-world geomodels. We then conducted a sensitivity
study of the topological spread with respect to the geomodel resolution. This allowed us to determine the appropriate geomodel
resolution necessary for our experiment. Next, we performed three separate simulations to compare different approaches:

1.  A Monte Carlo simulation of the prior uncertainty for $1000$ samples, to evaluate the spatial uncertainty and the topological
    spread of the resulting geomodel ensemble. This serves as our 'base case' uncertainty for comparison with the following
    two simulations.

2.  An ABC-REJ simulation using the initial geomodel topology graph (see Fig. 5b) to represent our geological knowledge.
We used an error threshold of $\epsilon = 0.025$ for $1000$ accepted posterior samples, as the threshold was small enough to
    constrain the posterior topology spread to the initial geomodel topology graph.

3.  An ABC-SMC simulation using the same initial geomodel topology graph. We ran six SMC epochs using $\epsilon$ values of
    $0.3$, $0.2$, $0.1$, $0.075$, $0.05$ and $0.025$. Each epoch was run for $1000$ accepted posterior samples.

## 3  Results

### 3.1  Synthetic Fault Model

Simulating the uncertainties encoded in the prior parameterization resulted in $100$ unique model topologies within the geomodel
ensemble of $2000$ models, with $18$ topology graphs occurring at least ten times and the most frequent $14$ making up $90\,\%$ of
geomodel ensemble topologies. It is also notable that the most frequent topology graph ($29.5\,\%$) is not the initial (mean
prior) topology graph ($15.6\,\%$), but rather represents models where the Shale layer (green) of the foot wall shares an across-
fault connection with the Sandstone 2 layer (red) of the hanging wall. The uncertainty of the prior geomodel ensemble is
visualized in Figure 6a-c in XZ-, YZ- and XY-sections as Shannon entropy, as described in the methodology. All three sections
through the model show clearly the uncertainty of the layer interface position and highest uncertainty around the fault surface.
In comparison, applying a single topology graph as a summary statistics to the simulation using ABC leads to significantly
reduced uncertainty throughout the geomodel ensemble (see Fig. 6d-f), with average geomodel ensemble entropy being reduced
from $\overline{H}_{prior} = 0.44$ down to $\overline{H}_{posterior} = 0.31$, a drop in geomodel uncertainty of nearly $30\,\%$. Visualizing the entropy

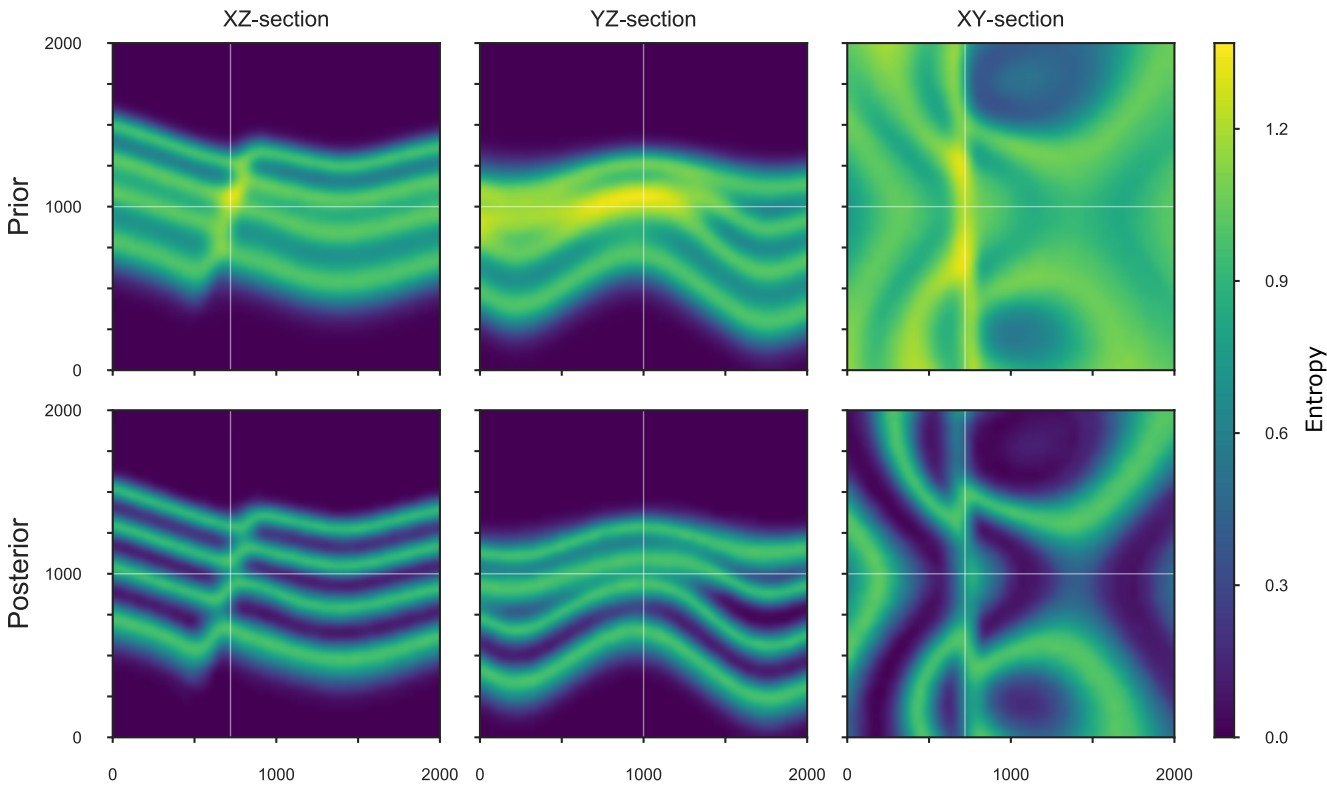

**Figure 6.** Shannon entropy slices in the XZ- (left), YX- (center) and XY-plane of the prior (top) and posterior (bottom) geomodel ensemble.

difference between the prior and the posterior geomodel ensembles shows the highest reduction in entropy for the two inner layer interfaces (see Fig. 7), and not around the fault surface. As expected, constraining the simulation using a single topology graph with an error of $\epsilon = 0$ collapses the number of geomodel ensemble topologies from 100 down to 1.

Figure 8 plots histograms and their kernel density estimations (KDE) of the input parameter distributions of prior (grey) and posterior (coloured) samples. The strongest change in mean from prior to posterior distributions occurred for the vertical interface location perturbance priors of Sandstone 2 (red), Shale (green) and Sandstone 1 (brown; see Fig. 8), with the first shifted to higher mean z-values and the latter two shifted deeper by $-72\,m$ and $-53\,m$, respectively. Additionally, the initially normally distributed prior of Sandstone 1 shows a strong negative skewness of $-0.61$ in the posterior distribution. Standard deviation for the Siltstone and Shale interface distributions was reduced by roughly $32\,\%$ and $40\,\%$ respectively. The prior and posterior distributions for the lateral and vertical fault parameter uncertainties show no significant difference (e and f).

### 3.2 Case Study: The Gullfaks Field

Forward simulation of the prior uncertainties of the Gullfaks geomodel resulted in 676 unique geomodel topologies within a 1000-model ensemble, with 116 unique topologies occurring more than once. Again, the most frequent topology graph is




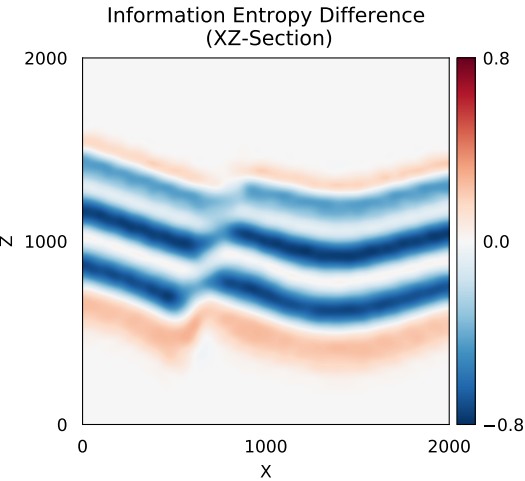

**Figure 7.** XY-Section of entropy difference between the forward simulated entropy and the approximate posterior entropy. The plot highlights areas where the entropy was reduced (blue), increased (red) and kept constant (white).

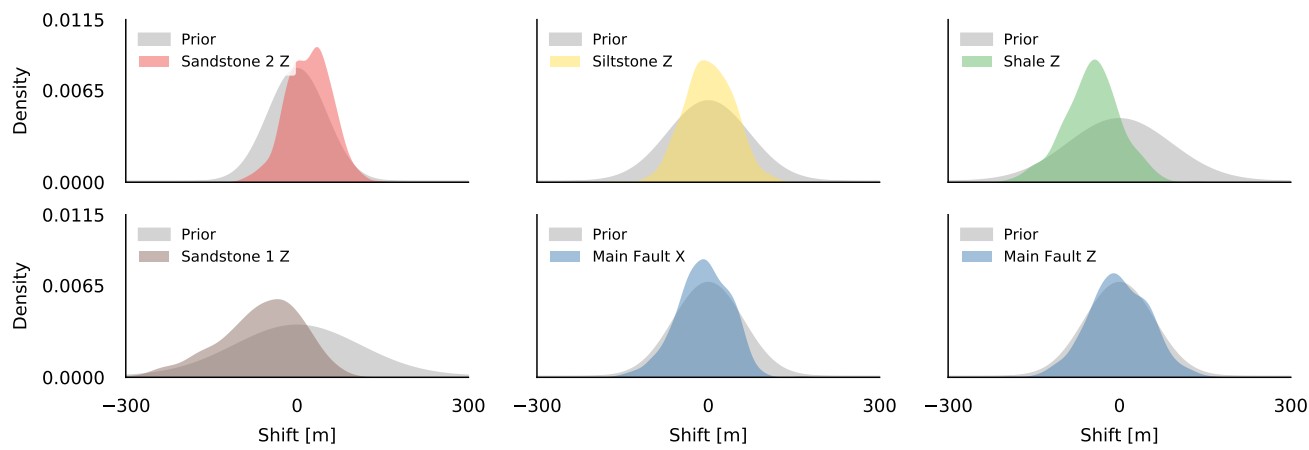

**Figure 8.** Prior (grey) and posterior (color) kernel density estimations for the different stochastic model parameters for our synthetic fault model.

not the initial (mean prior) topology graph. The uncertainty of a XZ-section of the forward ensemble is visualized in Figure 290 10a using Shannon entropy. The section illustrates the general trend of uncertainty throughout the forward simulation: we observe highest uncertainty surrounding the two faults in the geomodel, especially around the eastern fault. The area also shows increased uncertainty due to the interaction of layer interfaces, the fault and the vertical vicinity of the BCU.





Applying the initial topology graph as a constraining summary statistics using ABC with rejection sampling (ABC-REJ) using a threshold of $\epsilon = 0.025$ (chosen empirically), results in much reduced uncertainty, as exemplified by the entropy section shown in Figure 10b. At this threshold, the approximate posterior geomodel ensemble contains only the applied initial topology graph. Using rejection sampling with such a strict threshold resulted in a very low acceptance rate of $0.0059$, which required about 40 hours of simulation time to obtain 1000 posterior samples[3]. In contrast, using a Sequential Monte Carlo sampling scheme (ABC-SMC) required only 3.96 hours to obtain the same number of posterior samples at the same threshold—a speed-up of 10.1. This includes the five sampling epochs using $\epsilon = \{0.3, 0.2, 0.1, 0.075, 0.05\}$ with 1000 accepted samples each, used to sequentially adapt the priors.

Figure 12a shows the number of unique topologies for forward simulations and each threshold of the ABC-SMC. As we iteratively lower the acceptable threshold during the SMC simulation, the simulated and accepted topologies iteratively converge towards the topology graph we used as our prior geological knowledge. The average geomodel ensemble entropy $\bar{H}$ is also iteratively decreasing from $0.233$ for the forward simulation down to $0.112$ at $\epsilon = 0.025$ (see Fig. 12b), showing how fixing a probabilistic geomodel to a single topology graph can significantly reduce, or rather significantly constrain, the simulated uncertainty.

Figure 9 shows how the ABC-SMC simulation iteratively affects the probability distributions of selected probabilistic geomodel parameters with decreasing thresholds $\epsilon$. Each row shows the consecutive epochs of the ABC-SMC simulation and corresponds to a specific $\epsilon$. Each column describes a different stochastic parameter in the stochastic model. By applying the initial topology graph of the geomodel as our summary statistics, we can directly see here how the parameter distribution for the BCU (Fig. 9a) shifts its mean $\mu$ by $47.4\ m$ upwards and reduces its standard deviation $\sigma$ by $35.8\ \%$ to accommodate our geological knowledge about the geomodel topology. We can observe this effect in the entropy section of the posterior geomodel ensemble as well (Fig. 10b). In Figure 11, we show the difference in entropy between the prior and approximate posterior geomodel ensemble shown in Figure 10, where areas with decreasing entropy values are shown in blue, increasing values in red. We observe here how the BCU moves upward and increases the entropy there, while lowering entropy in the lithologies below. The parameter distributions for Tarbert B (Fig. 9b, red) and Etive B (Fig. 9c, green) show similar behaviour: shifted mean and reduced standard deviation to accommodate the topology information. We see a much stronger reduction in standard deviation for the two faults (Fig. 9d,e): $80.4\ \%$ and $80.0\ \%$ for Fault A and Fault B, respectively. This is also shown as the strongest reduction in entropy in Figure 11.

## 4 Discussion

We showed how topology information, as an encoding for important aspects of geological knowledge and reasoning, can be included in probabilistic geomodeling methods in a Bayesian framework. The simulation experiments for our two case studies demonstrated that we are able to approximate posterior distributions to obtain probabilistic geomodel ensembles that honour

---

[3]The experiment was run on consumer-grade hardware and leveraging GPU computation: Intel Core i5-8600K @ 3.60GHz, Nvidia GeForce RTX 2070 8GB GDDR6, 16 GB DDR4 RAM @ 2133MHz.





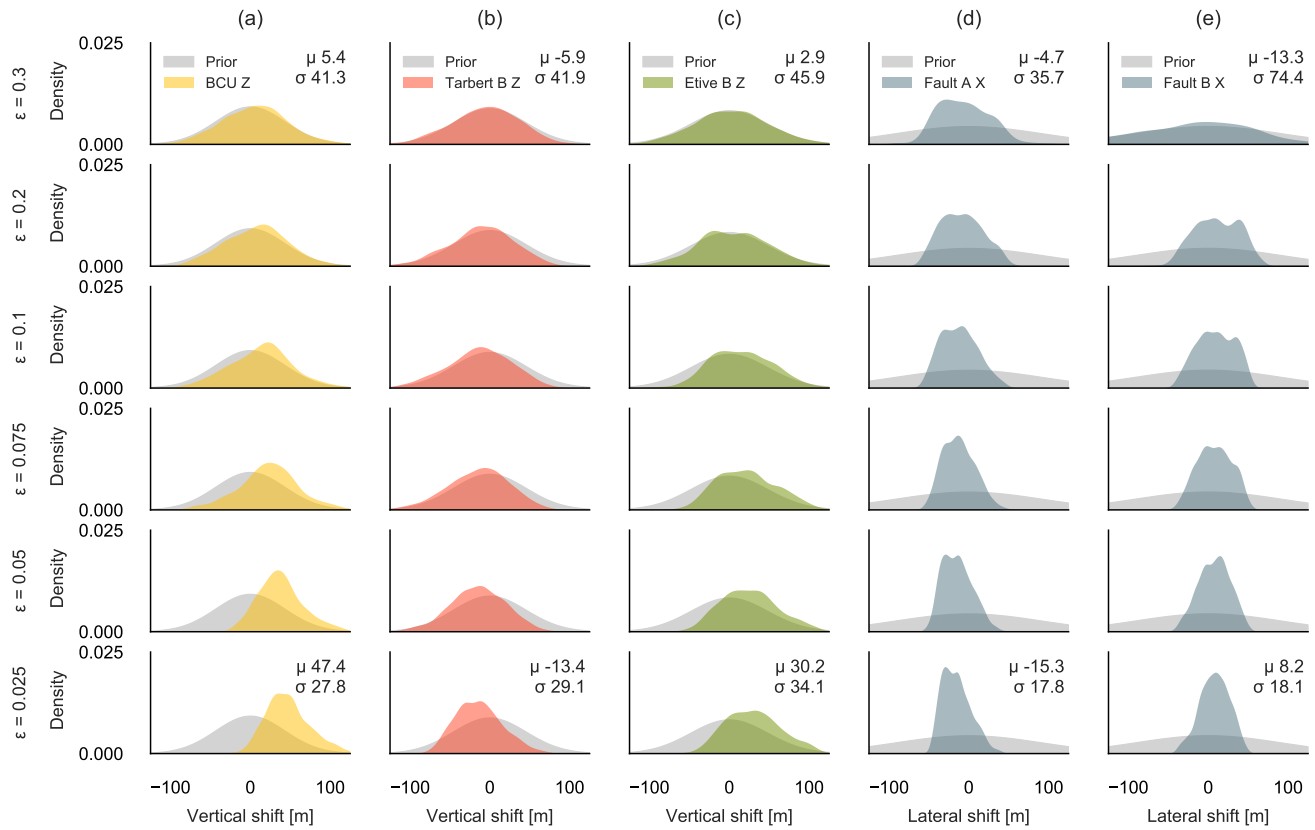

**Figure 9.** Prior (grey) and posterior (colored) kernel density estimations for selected model parameters (a-e) for the 6 epochs (each row represents an epoch) of the ABC-SMC simulation of the Gullfaks case study, showing how the simulation iteratively approaches the approximate posterior distribution, which shows the possible parameter uncertainty given our topological information. Mean $\mu$ and standard deviation $\sigma$ shown for the first and last epochs.

both our prior parameter knowledge and qualitative geological knowledge. If the applied topological information is meaningful,

then the constrained stochastic geomodel ensemble will see a meaningful reduction in uncertainty, and will subsequently allow for more precise model-based estimates and decision-making (Stamm et al., 2019). More importantly, the (approximate) Bayesian approach requires the explicit statement of the geological knowledge (here the topology information) used in the probabilistic geomodel, increasing the transparency of assumptions made during the geomodeling process and any subsequent decisions.

With our approach, we directly address a scientific challenge raised in recent work by Thiele et al. (2016b), that known topological relationships are frequently not honoured during the probabilistic modeling process, thus potentially invalidating large parts of the resulting geomodel ensemble. Injecting topology information into a Bayesian approach allows us to obtain topologically valid, and hence geologically reasonable, geomodel ensembles. And, although we have only used simple topology





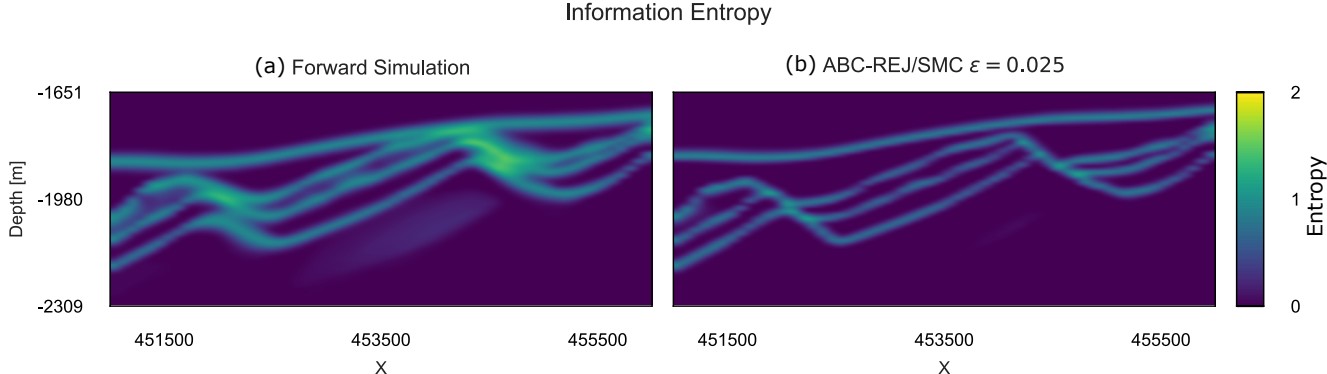

**Figure 10.** (a) Section of the entropy block of the forward simulation for the prior uncertainty ($H_T = 0.223$); (b) Section of the entropy block of the final epoch ($\epsilon = 0.025$) of the ABC-SMC simulation ($H_T = 0.113$).

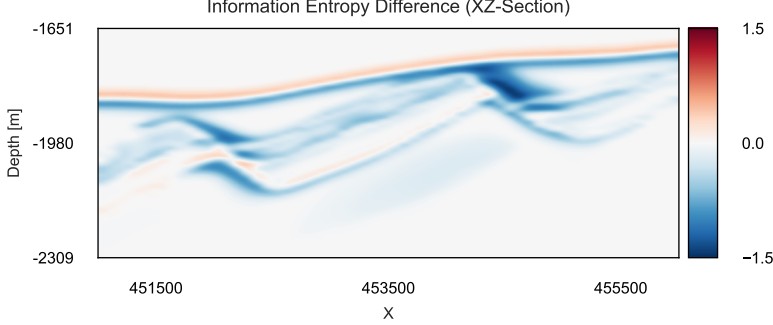

**Figure 11.** XZ-Section of entropy difference between the forward simulated entropy and the approximate posterior entropy $H$ ($\epsilon = 0.025$). The plot highlights areas where the entropy was reduced (blue), increased (red) and kept constant (white).

information within this study, the demonstrated ABC approach allows to easily scale the amount of topology information used:

from simple True-False comparisons of single topology graphs to the use of a whole range of topology graphs and relationships.

The work of Pakyuz-Charrier et al. (2019) shows how clustering of probabilistic geomodel topologies can be used to differentiate between different modes of topologies. Their approach compares geomodel topologies by describing them as half-vectorized adjacency matrices, resulting in a binary string that can be compared using the Hamming distance (Hamming, 1950). It could be considered as a different distance metric in the ABC approach presented in this work to constrain the

340 simulated probabilistic geomodel. And, while their work focuses on the analysis of existing probabilistic geomodel ensembles, our approach focuses on learning probabilistic geomodels on topology information while reducing the number of required iterations through use of advanced sampling techniques.



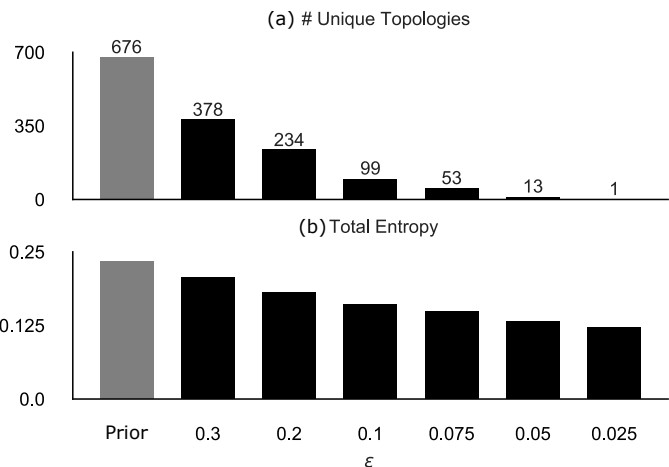

**Figure 12.** (a) Number of unique topologies within the geomodel ensembles of each SMC epoch, showing the iterative reduction in topological uncertainty throughout the SMC simulation; (b) Average geomodel entropy of the ensembles for each epoch, showing how the reduction of topological uncertainty shown in (a) affects the total geomodel uncertainty.

As more complex geomodels strongly increase the required parametrization to accurately describe the model domain in a probabilistic framework, constraining them with topological information could help keep this parametrization at computationally
feasible levels by reducing the parameter dimensionality, while still obtaining meaningful geomodels (e.g. free of modeling artefacts caused by random perturbations of the limited input data). This would not work using an inefficient rejection sampling scheme (e.g. ABC-REJ), but would rather require the use of "adaptive" sampling algorithms to efficiently explore the posterior parameter space without wasting too much computing power on rejected models (e.g. ABC-SMC). In our Gullfaks case study, we have not only shown the efficacy of the method on a real-world example, but demonstrated the stark increase in
computational efficiency when using advanced sampling techniques. The SMC sampler used in our work requires manual setting of the acceptance thresholds, which directly influence the algorithm's efficiency in acquiring samples of the approximate posterior distribution. Adaptive SMC methods automatically tune acceptance thresholds to increase sampling efficiency on-the-fly to minimize computation time and avoid manual (subjective) selection of thresholds (Del Moral et al., 2012).

Sadegh and Vrugt (2014) describe a more complex ABC algorithm based on Differential Evolution Adaptive Metropolis
(DREAM-ABC) and demonstrate its much higher efficiency in approximating the posterior. It might be of particular interest for the approximate inference of complex structural geomodels with topology constraints, as it has shown promise to very efficiently explore high-dimensional (read: large amount of prior parameters) and multi-modal parameter spaces. When using multiple topology graphs (which are discrete) in an ABC framework, the posterior parameter space may potentially become multi-modal, which poses significant challenges for traditional Markov Chain-based samplers (Feroz and Hobson, 2008).
The approach by Sadegh and Vrugt (2014) is based on combining multiple Markov chains, which natively supports parallel computing and would thus allow for a high scalability of the approach to complex, computationally intensive geomodels.





Alternatively, Bayesian Optimization for likelihood-free inference (BOLFI; Gutmann and Corander, 2016) could be worth considering for complex structural geomodels. The method abstracts the simulator/implicit function into a statistical surrogate model between the priors and the summary statistics and then attempts to minimize their distance, with the potential to significantly reduce the number of needed computations of the geomodel. Overall, the spatial and discrete nature of geomodels and the use of discrete summary statistics poses unique challenges to sampling algorithms, requiring further research to identify algorithms that can confidently converge and minimize the high computational cost of probabilistic 3-D geomodels.

The method demonstrated the effect of topology information on geomodel uncertainty—showing how well the parametrization of a probabilistic geomodel fits our geological assumptions. The acceptance rates during sampling could potentially be used as a proxy for the validity of our assumptions: low acceptance rates could reveal a bad fit between our model and our added geological knowledge and reasoning. Using entropy-difference plots, the effect of geological assumptions on the uncertainty can be analysed spatially, e.g. how it reduces (or increases) around faults and other structures in the geomodel or other summary statistics of the geomodel, such as the gross rock volume of a potential reservoir across all fault blocks (or compartments) of interest.

## Summary

- We have shown how to use Approximate Bayesian Computation to constrain probabilistic geomodels so that the approximate posterior incorporates topology information.

- The method enables additional geological knowledge and reasoning to be explicitly encoded and incorporated into probabilistic geomodel ensembles, potentially increasing transparency of the modeling assumptions.

- As opposed to standard MC with rejection, the implemented SMC approach makes the use of ABC feasible in realistic settings. Further research into using more advanced sampling schemes could provide additional speed-ups in obtaining the posterior geomodel ensemble, which is especially relevant for computationally more expensive complex geomodels with large parametrizations.

**Table 1.** Distribution parameters for prior parametrization of the synthetic fault model.

| Name | Distribution | $\mu$ [m] | $\sigma$ [m] |
|---|---|---|---|
| Sandstone_2_Z | Normal | 0 | 50 |
| Siltstone_Z | Normal | 0 | 70 |
| Shale_Z | Normal | 0 | 90 |
| Sandstone_1_Z | Normal | 0 | 110 |
| Main_Fault_X | Normal | 0 | 60 |
| Main_Fault_Z | Normal | 0 | 60 |



**Table 2.** Distribution parameters for prior parametrization of the Gullfaks case study.

| Name | Distribution | $\mu$ [m] | $\sigma$ [m] |
|------|--------------|-----------|--------------|
| BCU Z | Normal | 0 | 43.3 |
| fault3 X | Normal | 0 | 90.9 |
| fault4 X | Normal | 0 | 90.5 |
| tarbert A Z | Normal | 0 | 46.5 |
| tarbert B Z | Normal | 0 | 45.5 |
| tarbert C Z | Normal | 0 | 44.2 |
| ness A Z | Normal | 0 | 48.6 |
| ness B Z | Normal | 0 | 46.7 |
| ness C Z | Normal | 0 | 45.1 |
| etive A Z | Normal | 0 | 50.9 |
| etive B Z | Normal | 0 | 48.1 |
| etive C Z | Normal | 0 | 46.3 |

*Competing interests.* The authors declare that they have no conflict of interest.

*Disclaimer.* This research was conducted within the scope of a Total E&P UK-funded postgraduate research project.

*Acknowledgements.* We would like to thank Total E&P UK in Aberdeen for funding this research. We also thank Fabian Stamm for providing the wonderful synthetic geomodel used in this paper.

*Code and data availability.* Input data and scripts to run the model and produce the plots for all the simulations presented in this paper are archived at Zenodo (Schaaf, 2020). GemPy 2.1 can be accesses via the published releases on the official GitHub repository at github.com/cgre-
390 aachen/gempy or alternatively at Zenodo (de la Varga, 2020)





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
