# Peer review of "Constraining stochastic 3-D structural geological models with topology information using Approximate Bayesian Computation in GemPy 2.1"

_Geoscientific Model Development, 2020_

## Referee Comment (RC1) · Ashton Krajnovich (Referee) · 17 Oct 2020

General comments:

The article "Constraining stochastic 3-D structural geological models with topology information using Approximate Bayesian Computation using GemPy 2.1" provides a practical and easy to understand method for the use of topological analysis in probabilistic geomodeling as part of a likelihood-free Bayesian inference scheme. The article moves research in probabilistic geomodeling in a meaningful direction towards the incorporation of geologic knowledge (or "knowledge-based inputs"), which is a valuable contribution as the uncertainty of geological knowledge is a traditionally underrepresented aspect of geologic modeling. The article does so by building upon recent works in topological analysis of 3D geologic models, demonstrating the application of geologic knowledge in the form of topology graphs describing the known distribution and relationships of normally-faulted stratigraphic units. The method put forth in the article presents practical advancements to enable the broader use of knowledge-based inputs in probabilistic geomodeling through the combined use of likelihood-free Bayesian inference (via Approximate Bayesian Computation (ABC)) and advanced sampling schemes (Sequential Monte Carlo (SMC)). The positive implications of the use of these tools in the research are clearly stated: circumventing the intractability of defining mathematical likelihood functions for abstract geologic knowledge and demonstrating processing performance improvements through a brief discussion of simulation efficiency. The authors give clear credit to related works both in the realm of geologic modeling as well as the broader fields of topological analysis and Bayesian statistics. The work fits well into the current state of probabilistic geomodeling research, with the research objectives achieved coinciding with recommendations made in recent works. The work provides necessary codes, algorithm descriptions and parameter files for reproduction of the research results both in the text and as supplemental material (Zenodo DOI). The article title effectively communicates the contents of the paper.

The work utilizes the developing, open source GemPy geologic modeling environment effectively, particularly highlighting the strengths provided from GemPy's ability to efficiently integrate stochastic simulation, topological analysis and 3-D geomodeling in a single platform. This integration is a critical component necessary to improve geomodel processing efficiency through iterative sampling schemes such as SMC. The discussion of the processing efficiency improvements however is brief, and as a significant contribution of this work, should be addressed in more detail (see specific comments for suggestions).

A key assumption that the demonstrated method operates under is that the observed topology graph of the geologic model being analyzed is known without uncertainty. This assumption needs to be stated more clearly to the reader, as the current presentation is confusing (e.g., the train of thought from line 21-24 is not in line with the proposed method in line 64). This is important as the convergence to a single topology graph plays a role in the significant reduction in uncertainty seen across the final probabilistic geomodel ensembles, so the discussion of this reduction should be clearly stated in light of the use of a known subsurface topology. Clarifying this assumption could also help clarify confusing mathematical notations and technical terminology used when describing summary statistics in Section 2.3.2.

Another aspect of the paper that is not sufficiently addressed is the selection and definition of the input parameter prior uncertainty distributions. It is unclear whether they were defined as broad, non-informative priors, based on empirical analyses, drawn from previous works, or assumed by the modeler. While the focus of the paper is on the use of ABC to incorporate geologic knowledge in the form of topology information, the core methodology is based in input-based, probabilistic geomodeling, and as such the discussion of input parameter prior uncertainty distribution selection and characterization needs to be discussed in some more detail.

The structure, language and mathematical notation of the paper could use some improvement. Figure callouts are often out of order and separated from their referenced figures by up to a page or more, hurting the paper's readability. Some figures might also be combined for ease of reference, e.g., Figures 6 and 7 and Figures 10 and 11. The synthetic and realistic geologic models case studies share many similarities (differing mainly in size and the presence of an overlying unconformity), leading to some avoidable repetition of information in the description of methodology and results between these two models. The language used leans towards a somewhat casual style, exhibiting some repetitive sentence structures and, in some cases, run-on sentences and other grammar related readability issues (see technical corrections for edits and

suggestions!). The mathematical notation is unclear in some places (e.g., Paragraph at line 161) and should be reviewed to be consistent and in line with the general statistical literature (rather than just from a specific cited work). Some technical terms lack definitions before their introduction (line 21) and in a few cases are provided with confusing definitions (e.g., Paragraph at line 161). These issues do not significantly impede the quality of the research, but will require minor revisions.

Overall, this research is a valuable and fitting contribution to GMD. Minor revisions are suggested regarding structure, figures, language, mathematical notation and definition of technical terms. More importantly, additional clarification and discussion is necessary regarding: (i) the reasoning behind and implications of key assumptions used in the work, namely the use of a known topology graph, and (ii) on the description of input parameter prior uncertainty distributions (and their impact on potentially low model acceptance rates). Following these revisions and clarifications, I would recommend this paper for publication in GMD.

Specific Comments:

Title: Consider rephrasing to avoid the repetitive use of the word "using". I would suggest: "Constraining stochastic 3-D structural geological models with topology information using Approximate Bayesian Computation in GemPy 2.1"

Abstract: As the research is built in the GemPy environment, it would be beneficial to highlight it's usage in the abstract (perhaps at Line 13).

Line 129: Sentence requires revision to be accurate about what the likelihood function represents in Bayes' theorem. I suggest: "This updating process relies on the use of a likelihood function p(y|theta), representing the conditional probability of the observed data y given the prior probability of the underlying parameter theta and the theoretical connection to the occurring event."

Line 144: You have reversed the conditional probability described by the likelihood

function, which is: the likelihood for observing the data y, given the model based on uncertain parameters theta.

Line 147: This is unclear, as likelihood functions are inherently encoding information regarding not just the parameters theta, but also the observations y and the assumed theoretical relationship between theta and y. Consider removing or revising.

Section 2.3.2: This section requires additional clarification between "observed data" and "simulated data". Refer to the treatment of ABC in Gelman et al., 2004 where y is the observed data (observed "summary statistic" in ABC) and y-rep is the simulated data (simulated "summary statistic" in ABC). The use of y-hat to represent the observed summary statistic and y to represent the simulated summary statistic creates additional confusion (as the observed data introduced in Bayes' theorem were defined as y, not y-hat).

Line 156-157: Perhaps add a reference to (Wood and Curtis, 2004)? (Geological prior information, and its applications to geoscientific problems)

Line 160: Please add an additional clarifying sentence on what the summary statistic is in this work rather than the short parenthetical (to avoid confusion with typical summary statistics like mean, mode, median etc.). Also, a comment: In the proposed (approximate) inference scheme, the new evidence y (or data) is the "summary statistic". So, while the definition of the additional term "summary statistic" to describe "y" is useful for highlighting the approximate nature of ABC, the equivalency of these two terms should be clarified for the reader.

Line 162-163: Clarify the 2nd part of the sentence to illustrate that the "observed summary statistic y-hat" is static for the entire geomodel ensemble (i.e., the known, observed topology graph), while "the summary statistic y" is tied to each individual geomodel realization (i.e., a simulated topology graph).

Line 165: Theta-prime has not been introduced. What does it refer to as opposed

to theta? I assume you are referring to a single draw from the parameter distribution theta, but please clarify. When relying on mathematical notations from another work (the ones in question here seem to be borrowed from Sadegh and Vrugt, 2014), make sure notations are introduced properly. It also helps to also have a "sanity check" to make sure that the notation used is not confusing with respect to the broader statistical literature (e.g., where the observed data in Bayes theorem are typically represented without a ˆ or ')

Section 2.5: Section could be made much more concise to avoid excessive overlap with existing works (seeing as the major contributions of the paper are not focused on novel applications of Shannon entropy).

Line 227: How and why were the prior uncertainty ranges chosen? Were they considered to be broad, non-informative priors, derived emprically, based on background information or simply assumed by the modeler for the sake of simulation? Same question should be addressed more directly for the Gulfalks case study as well (Line 249-251), where the uncertainties appear to be derived from the referenced work though this is not stated definitively. Also, just a comment: I am quite interested to see how incorporating structural uncertainty (by way of the methods put forth by Pakyuz-Charrier et al., 2018a,b, Roberts et al., 2019 or Krajnovich et al., 2020) would influence the geomodel topology.. Intuitively, there is a high potential for confounding effects on the range of possible geomodel topologies when interface location and interface/fault orientation are varied together!

Line 246: How was the interface uncertainty applied to the surface points? Independently at each node, or generally to the set of surface points (so as to retain surface shape). From reading into the supplemental codes, it appears that the uncertainty was applied to the group of surface points – but this information needs to also be included in the text for the typical reader. This also applies to the synthetic model, which appears (from the code provided) to have been modeled from similar groups of surface points, though this is not clarified in the text.

Line 251: Tying back to the earlier comments on how prior uncertainty ranges were chosen, I believe that "ease of implementation" is somewhat of an inconclusive reasoning. The rest of the sentence provides more meaningful perspective but still could be expanded upon (e.g., what is "simplified uncertainty modeling" in this context?). Please add some more detail.

Line 268: A figure representing this most frequent topology graph from simulation (or other selected simulated topology graphs) would be quite insightful, especially if accompanied by a discussion of their geologic significance (e.g., tying back to points made during the introduction (Line 51), did any simulated topology graphs represent a compressional rather than extensional tectonic regime?). If length permits of course - perhaps if some figures are combined or suggested section lengths reduced, this could be added.

Line 287: Since the Jaccard Index used could allow for multiple topologies to be present in the final model ensemble (depending on the rejection threshold used), it would be beneficial to see some exploration of what these possible model topologies looked like (how geologically unrealistic do they get? Are all 675 unused topologies absolutely unrealistic?). Including a discussion of this sort would help guide future works investigating uncertainty of the applied topology information itself (without requiring reproducing the results to show geomodel uncertainty when multiple simulated topologies were present in the final ensemble). See also Comment for Line 268.

Line 295: In line with the missing clarification regarding the assumption of the observed topology graph being known without uncertainty, add some clarification behind the reasoning for setting the rejection threshold such that only the applied initial topology remains in the probabilistic geomodel ensemble. Was the goal of empirical testing of thresholds to find the largest threshold which resulted in only a single model topology remaining across the probabilistic geomodel ensemble?

Line 297: How does simulation time for ABC-REJ compare to simulation time for the

standard MC approach?

Line 298: This is a significant improvement in efficiency! Perhaps include a description of acceptance rates from each epoch of SMC, or at least a comparison of the final acceptance rate at the threshold value of 0.025 in SMC for comparison with the rate given for REJ. This information might fit naturally in Figure 12.

Line 324: If the information applied were non-meaningful (e.g., an incorrect topology graph), the geomodel ensemble would likely still exhibit a reduction in entropy due simply to the convergence of the model realizations towards the single model topology applied. That is, the reduction in uncertainty is arising from the reduction of possible model topologies, not necessarily the meaningfulness of the model topology used in the ABC algorithm.

Line 334: It appears that expanding the ABC approach proposed here to incorporate multiple observed topology graphs would not be a matter of "easily scaling". Revise to clarify that the general ABC framework would definitely allow for this, although it would require reparameterizing the current summary statistic and discrepancy measure (distance function), and also possibly changing the simulation method (as mentioned in Line 357-359).

Line 335: This would be a good place to bring up again the implications of using the demonstrated ABC approach if there were uncertainty about the observed topology graph.

Line 345: "…reducing the parameter dimensionality" – how so? The number of input parameter probability distributions is the same in standard MC or in ABC-REJ/SMC. The computation efficiency improvements arrive from reducing the number of input parameter draws that are run through uncertainty propagation to the 3D geologic model space, which in SMC also allows for reducing the size of the uncertainty space (note, not the parameter dimensionality) iteratively.

Line 370: This was not discussed earlier in Section 3.2 when the acceptance rate was initially 0.0059 (0.59%). Does that low acceptance rate warrant reassessing the prior input uncertainties used in the probabilistic geomodeling? Should be discussed to better frame the current work and guide future work.

Figure 4: Consider replacing X & Y with N & E to be more intuitive for geoscientists. Applies to all figures of geomodels with labeled axes.

Figure 6: In my opinion, the XZ difference section (and possibly then also XY and YZ sections) from Figure 7 could be appended onto Figure 6 for ease of reference. Also, what do the overlain crosshairs show?

Figure 8: Figure does not show (a), (b), (c)... tags. Also, as mentioned in the comment for Line 279, the figure does not show histograms.

Figure 10: The significant reduction in model entropy indicates the strong dependence on the initial topology used - this potential source of bias should be addressed. Please discuss the implications of using a rejection threshold which only allows one model topology across the entire final ensemble of geomodel realizations. Since the authors are operating under this (valid) assumption, it needs to be clearly stated earlier that the initial geomodel topology is "known" and treated without uncertainty. See also comments regarding Lines 324, 295, 287 and 268. Also, I believe this figure could be merged with Figure 11.

Figure 12: Figure needs correction to show Y-axes labels. Perhaps acceptance rates per epoch would be useful to add as well, as they are tied to the processing efficiency improvement of 10.1x (see comment regarding Line 298).

Technical Corrections:

See the annotated PDF provided as supplementary material.

Please also note the supplement to this comment:

https://gmd.copernicus.org/preprints/gmd-2020-136/gmd-2020-136-RC1-supplement.pdf

**Supplement:**

[revised manuscript text omitted]

---

## Referee Comment (RC2) · Anonymous Referee #2 · 21 Oct 2020

General Comments:

The article proposes a method for applying constraints derived from topological geological knowledge to ensembles of geomodels generated by Monte Carlo algorithms. The concept is simple, its application fits well with common Bayesian methodology and I can see its practical value and relevance to this journal. Experiment setup and results are well presented and the software used is thoroughly described so as to be reproducible. While some readers will have questions about how exactly various prior parameters were chosen, I interpret this article as being about the method of incorporating topological data and not the practice of deciding on priors which is covered in many other works. I am asking for minor revision for the following reasons:

1) Many sentences are poorly worded.

2) The term 'likelihood-free' and its implications are presented inconsistently and in a very misleading way.

3) Terms like observation, prior and likelihood are applied in an unclear manner often contradicting convention.

4) The method description is much longer and more elaborate than it needed to be. This stems from the choice to phrase the approach in terms of Approximate Bayesian Computation (ABC).

To be clear, I think the choice to frame this in terms of ABC was unnecessary and only makes the article much longer and more tedious than was needed. I am not asking for this to be changed, as that would be more effort than it is worth. I make this point here simply for the record and in the hope that it will help make future articles less convoluted.

ABC (both rejection and SMC) approximates likelihood in the following ways: Firstly, exact conformity to the the predictions of the model (theta) is relaxed using a distance measure and threshold. Secondly, if the simulation linking cause (theta) to observation (y) is stochastic then it uses a finite set of MC realisations instead of an integral over all outcomes of the random variables not of interest.

Neither of these properties are used in the work presented here. Nothing of what separates ABC from the traditional Bayesian approaches is used here. That is not to say that it cannot be cast into ABC terms but rather that it is ABC only in the most superficial sense. The proposed approach could simply have been presented as applying a probabilistic constraint to an ensemble using rejection sampling and an SMC variant of it. This would have saved readers from a lot of irrelevant reading.

The topological constraint is computed from an 'initial' adjacency graph which is not observed but derived subjectively from interpreted seismic data (for the real data case). In common Bayesian practice this constraint would therefore be called a prior or empirical prior (if you want to emphasise that some data did inform it), not a likelihood. The use of distance measures to defines such priors is common in traditional Bayesian methodology and needs no appeal to more recent trends to be explained or justified. I do not have any issue with the subjective nature of deriving this as it is an unavoidable part of all inference, Bayesian or frequentest. However, the author claims to circumvent specification of a 'likelihood function'. In this case there was no need, the implicit 'likelihood' (empirical prior constraint on topology) is simply a uniform distribution on all geomodels not conforming to a binary topological constraint centred on the initial graph. It could have been specified simply as a typical prior constraint used in any traditional Bayesian application without the need to force an ABC interpretation. This would also have made the implicit assumptions more transparent. In short, the method could simply be described as using rejection sampling (much older than ABC) to apply a uniform empirical prior (also much older than ABC) on topology graphs of geomodel ensembles.

Whether the volumes (L,F & T), or connectivity graph, or adjacency graph, or its Jaccard index are considered y or S(y) is completely arbitrary here as none of these are directly obeserve in the experiments. The ABC distinction between y and S(y) doesn't aid anything in this particular application.

A more concerning problem is the author's misrepresentation of what the term 'likelihood-free' entails. The claim that specification of a 'likelihood' is circumvented is not true. Nor does this particular application simplify the process of specifying this probabilistic constraint any more than would be the case for any typical definition of a prior over model parameters in geoscience. As mentioned before, the implicit 'likelihood' used here is trivial and easy to formulate. This might not be the case for many other applications of ABC but it is here. I don't believe that the author intended for this

to be interpreted as ABC reducing the need for subjective assumptions but due to how things are phrased, many readers will take away exactly that message.

Specific Comments:

1) Remove claims all of circumventing or simplifying specification of topological knowledge due to ABC, these are untrue for the constraint presented here. I have highlighted these in the attached annotated pdf along with more detailed comments for each. Lines 60, 150, 155, 165.

2) Remove all uses of the term 'likelihood-free'. There is no place in this article where its use helps clarify how the proposed approach works. Its only effect is as a potential source of misinterpretation. To avoid unneeded additional review rounds I am asking for complete removal and not fixing its use.

3) Do not refer to the initial connectivity graph as an observation. It is a semi-subjectice semi-empirically derived parameter to a subjectively chosen constraint family. Describe briefly how it is obtained and what the reasoning behind using the 'initial' graph was.

4) I am not asking that you replace 'likelihood' with 'prior' when referring to your topological constraint. Instead, please state somewhere that you choose to go with this label but that it could also be considered a prior or empirical prior and that the application of these terms is not always clear cut. State that you are simply treating the adjacency graph (y) as an observation.

5) Two topology distance measures are defined in section 2.4 which are never used. Since they are never discussed, analysed or compared, they serve no purpose. I suggest you remove them to simplify and shorten the already long paper, but feel free to ignore this suggestion.

6) Remove the mention of fuzzy sets in section 2.5, it is not relevant. Your posterior is probabilistic not fuzzy. It represents degrees of certainty concerning a single underlying truth, not degrees of membership to a category.

[Figure]

7) Several sentences need to be reworded for clarity or readability. I have highlighted these in the annotated pdf. Please try to address most of them.

Please also note the supplement to this comment:
https://gmd.copernicus.org/preprints/gmd-2020-136/gmd-2020-136-RC2-supplement.pdf

**Supplement:**

[revised manuscript text omitted]

---

## Author Comment (AC1) · 17 Nov 2020

Thank you very much for your constructive review of our manuscript. We have incorporated many of your suggested improvements into our manuscript and believe this has significantly improved its quality and readability.

Most importantly, we have added our reasoning behind the choice of prior parameters and our reasoning behind choosing a known topology graph as our constraint. Additionally, we have revised the mathematical notation throughout the paper to make

it consistent and in line with the cited literature and have edited so that definitions of technical terms appear before use of them. The placement of the figures will be subject to the final paper typesetting done by the journal and is thus not final in the current manuscript. We have addressed many of the suggested improvements to language, grammar and figure annotations to improve readability.

Please find all our detailed responses to your comments in the supplementary material, along with the change-tracked manuscript.

Please also note the supplement to this comment:
https://gmd.copernicus.org/preprints/gmd-2020-136/gmd-2020-136-AC1-supplement.zip

---

## Author Comment (AC2) · 17 Nov 2020

Thank you very much for your constructive review of our manuscript. We have incorporated many of your suggested improvements into our manuscript and believe this has significantly improved its quality and readability.

We have addressed many of the reviewer's suggestions to improve wording and grammar throughout the manuscript. We have mostly removed the term "likelihood-free" from the manuscript to not confuse the reader, except where we directly reference lit-

erature that uses this term. We have elaborated on our choice of prior parametrization, have improved the mathematical notation of the statistical methods used so that the reader can more clearly differentiate between what our priors are and what our constraint is (i.e. the ABC-equivalent to the likelihood). We have not significantly cut the ABC method description, as we believe it will be valuable to the geoscientific reader to read about it in the context of geological modeling.

Please find all our detailed responses to your comments in the supplementary material, along with the change-tracked manuscript.

Please also note the supplement to this comment:
https://gmd.copernicus.org/preprints/gmd-2020-136/gmd-2020-136-AC2-supplement.zip

---

## Author Response (AR1)

**Response to RC1 – Ahston Krajnovich**

**AR = Authors response**

Thank you very much for your constructive review of our manuscript. We have incorporated many of your suggested improvements into our manuscript and believe this has significantly improved its quality and readability.

Most importantly, we have added our reasoning behind the choice of prior parameters and our reasoning behind choosing a known topology graph as our constraint. Additionally, we have revised the mathematical notation throughout the paper to make it consistent and in line with the cited literature and have edited so that definitions of technical terms appear before use of them. The placement of the figures will be subject to the final paper typesetting done by the journal and is thus not final in the current manuscript. We have addressed many of the suggested improvements to language, grammar and figure annotations to improve readability.

Please find all our detailed responses to your comments in the supplementary material, along with both the revised and change-tracked manuscript.

**Specific Comments**

Title: Consider rephrasing to avoid the repetitive use of the word "using". I would suggest: "Constraining stochastic 3-D structural geological models with topology information using Approximate Bayesian Computation in GemPy 2.1"

**AR: We have changed the title.**

Abstract: As the research is built in the GemPy environment, it would be beneficial to highlight it's usage in the abstract (perhaps at Line 13).

**AR: We now mention GemPy in the abstract to improve clarity.**

Line 129: Sentence requires revision to be accurate about what the likelihood function represents in Bayes' theorem. I suggest: "This updating process relies on the use of a likelihood function p(y|theta), representing the conditional probability of the observed data y given the prior probability of the underlying parameter theta and the theoretical connection to the occurring event."

**AR: We have incorporated your suggestion into the manuscript to improve clarity for the reader.**

Line 144: You have reversed the conditional probability described by the likelihood function, which is: the likelihood for observing the data y, given the model based on uncertain parameters theta.

**AR: Thank you for pointing out this error, we have switched that around!**

Line 147: This is unclear, as likelihood functions are inherently encoding information regarding not just the parameters theta, but also the observations y and the assumed theoretical relationship between theta and y. Consider removing or revising.

**AR: We have removed the sentence to avoid confusing the reader.**

Section 2.3.2: This section requires additional clarification between "observed data"

and "simulated data". Refer to the treatment of ABC in Gelman et al., 2004 where y
is the observed data (observed "summary statistic" in ABC) and y-rep is the simulated
data (simulated "summary statistic" in ABC). The use of y-hat to represent the observed
summary statistic and y to represent the simulated summary statistic creates additional
confusion (as the observed data introduced in Bayes' theorem were defined as y, not
y-hat).

**AR: Thank you for pointing out this mistake. We have changed the notation to be in line with the
literature and our description of Bayes' theorem.**

Line 156-157: Perhaps add a reference to (Wood and Curtis, 2004)? (Geological prior
information, and its applications to geoscientific problems)

**AR: Added reference to provide the reader with additional literature to understand the issue of
specifying likelihood functions in geology.**

Line 160: Please add an additional clarifying sentence on what the summary statistic
is in this work rather than the short parenthetical (to avoid confusion with typical
summary statistics like mean, mode, median etc.). Also, a comment: In the proposed
(approximate) inference scheme, the new evidence y (or data) is the "summary statistic".
So, while the definition of the additional term "summary statistic" to describe "y"
is useful for highlighting the approximate nature of ABC, the equivalency of these two
terms should be clarified for the reader.

**AR: We added additional clarification on what is usually used as summary statistics, and why we
use topology graphs when comparing geomodels.**

**"While summary statistics are often measures such as the mean, mode or
median of a model, they tend to be meaningless in summarizing geomodels. In this
work we use the geomodel topology graph as a summary statistic of the geomodel
to provide a meaningful comparison between geomodels."**

Line 162-163: Clarify the 2nd part of the sentence to illustrate that the "observed
summary statistic y-hat" is static for the entire geomodel ensemble (i.e., the known,
observed topology graph), while "the summary statistic y" is tied to each individual
geomodel realization (i.e., a simulated topology graph).

**AR: We have fixed the mathematical notation from y to S(y) when referring to the summary
statistic. We think this fixes the problem of clarity in this sentence. While we keep the observed
topology graph static in this experiment, this is by no means necessary, as we discuss in the paper.
We have also improved our discussion of this.**

Line 165: Theta-prime has not been introduced. What does it refer to as opposed to theta? I assume
you are referring to a single draw from the parameter distribution
theta, but please clarify. When relying on mathematical notations from another work
(the ones in question here seem to be borrowed from Sadegh and Vrugt, 2014), make
sure notations are introduced properly. It also helps to also have a "sanity check" to
make sure that the notation used is not confusing with respect to the broader statistical
literature (e.g., where the observed data in Bayes theorem are typically represented
without a ˆ or ')

**AR: Theta has been introduced in the previous section as the model parameter distributions. We have added an explanation that theta prime is a sample from these distributions.**

Section 2.5: Section could be made much more concise to avoid excessive overlap with existing works (seeing as the major contributions of the paper are not focused on novel applications of Shannon entropy).

**AR: We have cut detailed explanations of the Shannon entropy and refer the reader to the relevant literature.**

Line 227: How and why were the prior uncertainty ranges chosen? Were they considered to be broad, non-informative priors, derived emprically, based on background information or simply assumed by the modeler for the sake of simulation? Same question should be addressed more directly for the Gulfalks case study as well (Line 249-251), where the uncertainties appear to be derived from the referenced work though this is not stated definitively. Also, just a comment: I am quite interested to see how incorporating structural uncertainty (by way of the methods put forth by Pakyuz-Charrier et al., 2018a,b, Roberts et al., 2019 or Krajnovich et al., 2020) would influence the geomodel topology.. Intuitively, there is a high potential for confounding effects on the range of possible geomodel topologies when interface location and interface/fault orientation are varied together!

**AR: We have added explanation on how we chose the prior parametrization. As this paper focusses on developing and showcasing a new methodology for constraining uncertain geomodels using topology graphs, prior parametrization has not been a focus of the experiments. We refer the reader to other works, as also mentioned by the reviewer.**

Line 246: How was the interface uncertainty applied to the surface points? Independently at each node, or generally to the set of surface points (so as to retain surface shape). From reading into the supplemental codes, it appears that the uncertainty was applied to the group of surface points – but this information needs to also be included in the text for the typical reader. This also applies to the synthetic model, which appears (from the code provided) to have been modeled from similar groups of surface points, though this is not clarified in the text.

**AR: We have added the missing description on how the interface point uncertainty has been applied in both the synthetic and the real-world examples.**

Line 251: Tying back to the earlier comments on how prior uncertainty ranges were chosen, I believe that "ease of implementation" is somewhat of an inconclusive reasoning. The rest of the sentence provides more meaningful perspective but still could be expanded upon (e.g., what is "simplified uncertainty modeling" in this context?). Please add some more detail.

**AR: We have addressed our inconclusive writing and now more clearly describe how the model is perturbed and why we chose this approach.**

Line 268: A figure representing this most frequent topology graph from simulation (or other selected simulated topology graphs) would be quite insightful, especially if accompanied by a discussion of their geologic significance (e.g., tying back to points made during the introduction (Line 51), did any simulated topology graphs represent a

compressional rather than extensional tectonic regime?). If length permits of course - perhaps if some figures are combined or suggested section lengths reduced, this could be added.

**AR: This could be beneficial, but we think it would distract the reader from the main message of the manuscript: the method itself. Analysing ensembles of topologies for geological setting would require either painstaking manual evaluation of every model or require more research into defining how extension settings can be detected reliably from topology graphs. This would indeed be a very interesting topic, but out of the scope of this research.**

Line 287: Since the Jaccard Index used could allow for multiple topologies to be present in the final model ensemble (depending on the rejection threshold used), it would be beneficial to see some exploration of what these possible model topologies looked like (how geologically unrealistic do they get? Are all 675 unused topologies absolutely unrealistic?). Including a discussion of this sort would help guide future works investigating uncertainty of the applied topology information itself (without requiring reproducing the results to show geomodel uncertainty when multiple simulated topologies were present in the final ensemble). See also Comment for Line 268.

**AR: Please see our answer to the previous comment.**

Line 295: In line with the missing clarification regarding the assumption of the observed topology graph being known without uncertainty, add some clarification behind the reasoning for setting the rejection threshold such that only the applied initial topology remains in the probabilistic geomodel ensemble. Was the goal of empirical testing of thresholds to find the largest threshold which resulted in only a single model topology remaining across the probabilistic geomodel ensemble?

**AR: Our aim was indeed to show how to constrain a stochastic geomodel to a topological state --- to allow for the reliable simulation of uncertainty within a single topology state (kind-of like a single conceptual model). We think that more research into how to identify, and thus compare and constrain with, geologically similar geomodels from topology graphs is needed to allow the meaningful relaxation of the error threshold.**

Line 297: How does simulation time for ABC-REJ compare to simulation time for the standard MC approach?

**AR: We don't believe this comparison is very meaningful, as this is highly dependant on the choice of error, summary statistic etc. That's why we chose to only compare the ABC-REJ with the ABC-SMC – as they are aiming for the same constrained outcome, while the Monte Carlo forward**

Line 298: This is a significant improvement in efficiency! Perhaps include a description of acceptance rates from each epoch of SMC, or at least a comparison of the final acceptance rate at the threshold value of 0.025 in SMC for comparison with the rate given for REJ. This information might fit naturally in Figure 12.

**AR: We think that acceptance rates for the multiple epochs of the ABC-SMC are difficult to compare to the single acceptance rate of ABC-REJ. One could compare the average, or weighted average, but we believe that the comparison of overall simulation time is more meaningful in this case.**

Line 324: If the information applied were non-meaningful (e.g., an incorrect topology graph), the geomodel ensemble would likely still exhibit a reduction in entropy due simply to the convergence of the model realizations towards the single model topology applied. That is, the reduction in uncertainty is arising from the reduction of possible model topologies, not necessarily the meaningfulness of the model topology used in the ABC algorithm.

**AR: We fully agree, which is why we highlight the fact that only a meaningful constraint can lead to a meaningful reduction in model uncertainty.**

Line 334: It appears that expanding the ABC approach proposed here to incorporate multiple observed topology graphs would not be a matter of "easily scaling". Revise to clarify that the general ABC framework would definitely allow for this, although it would require reparameterizing the current summary statistic and discrepancy measure (distance function), and also possibly changing the simulation method (as mentioned in Line 357-359).

**AR: Incorporating multiple observed topology graphs for comparison in the ABC framework would indeed scale easily, as the computational complexity would scale linearly. Thus, a doubling in comparisons would double the Jaccard index computation, which by itself is trivial in terms of computation cost for such small networks in comparison with the computation cost of generating the geomodel in the first place. In the current implementation it would simply requiring looping over a set of topologies and computing the Jaccard index and accepting if one of them is below the allowed error threshold.**

Line 335: This would be a good place to bring up again the implications of using the demonstrated ABC approach if there were uncertainty about the observed topology graph.
**AR: The uncertainty in observed topology can be addressed via proposing several acceptable topologies or by scaling the error threshold.**

Line 345: ": : :reducing the parameter dimensionality" – how so? The number of input parameter probability distributions is the same in standard MC or in ABC-REJ/SMC. The computation efficiency improvements arrive from reducing the number of input parameter draws that are run through uncertainty propagation to the 3D geologic model space, which in SMC also allows for reducing the size of the uncertainty space (note, not the parameter dimensionality) iteratively.

**AR: We describe here a trade-off that could be made between uncertain geomodel parametrization and probability of subsequently simulated geomodel samples being valid geological models. Increasing geomodel complexity generally requires increased geomodel and statistical model parametrization – which thus scales the parameter space exponentially (see e.g. Betancourt, Michael. "A conceptual introduction to Hamiltonian Monte Carlo" (2017).). This increase in parameter space will drastically increase computational time. Thus, a balance generally needs to be made between model complexity and model parametrization. Our method could allow for lower levels of model parametrization while retaining model complexity by essentially filtering topologically wrong samples (which will inherently become more numerous when stochastically perturbing an under-parametrized complex geomodel).**

Line 370: This was not discussed earlier in Section 3.2 when the acceptance rate was initially 0.0059 (0.59%). Does that low acceptance rate warrant reassessing the

prior input uncertainties used in the probabilistic geomodeling? Should be discussed to better frame the current work and guide future work.

**AR: It might! It definitely warrants a good look at the prior parametrization. But stochastic geomodels have such inherent topological complexity, as minor changes in the location of interfaces across a fault can have various effects on model topology.**

Figure 4: Consider replacing X & Y with N & E to be more intuitive for geoscientists. Applies to all figures of geomodels with labeled axes.

**AR: We have chosen non-descript axis labels for these figures as this model is entirely synthetic.**

Figure 6: In my opinion, the XZ difference section (and possibly then also XY and YZ sections) from Figure 7 could be appended onto Figure 6 for ease of reference. Also, what do the overlain crosshairs show?

**AR: The lines show the locations of the respective other sections. We have added an explanation to the figure description to make this clearer.**

Figure 8: Figure does not show (a), (b), (c): : : tags. Also, as mentioned in the comment for Line 279, the figure does not show histograms.

**AR: We have added subfigure labels a-f and we have removed the outdated reference to histograms to only KDEs.**

Figure 10: The significant reduction in model entropy indicates the strong dependence on the initial topology used - this potential source of bias should be addressed. Please discuss the implications of using a rejection threshold which only allows one model topology across the entire final ensemble of geomodel realizations. Since the authors are operating under this (valid) assumption, it needs to be clearly stated earlier that the initial geomodel topology is "known" and treated without uncertainty. See also comments regarding Lines 324, 295, 287 and 268. Also, I believe this figure could be merged with Figure 11.

**AR: We have added explanation of this bias in our elaboration on why we chose to constrain with a single topology state in our method description. We hope this clears things up for the reader.**

Figure 12: Figure needs correction to show Y-axes labels. Perhaps acceptance rates per epoch would be useful to add as well, as they are tied to the processing efficiency improvement of 10.1x (see comment regarding Line 298).

**AR: We have added Y-axes labels and removed then redundant titles. We are not sure how acceptance rate would improve would improve the bar chart. We have chosen not to add it to keep the figure concise.**

**Response to RC2 – Anonymous Reviewer**

**AR = Authors response**

Thank you very much for your constructive review of our manuscript. We have incorporated many of your suggested improvements into our manuscript and believe this has significantly improved its quality and readability.

We have addressed many of the reviewer's suggestions to improve wording and grammar throughout the manuscript. We have mostly removed the term "likelihood-free" from the manuscript to not confuse the reader, except where we directly reference literature that uses this term. We have elaborated on our choice of prior parametrization, have improved the mathematical notation of the statistical methods used so that the reader can more clearly differentiate between what our priors are and what our constraint is (i.e. the ABC-equivalent to the likelihood). We have not significantly cut the ABC method description, as we believe it will be valuable to the geoscientific reader to read about it in the context of geological modeling.

Please find all our detailed responses to your comments in the supplementary material, along with both the revised and change-tracked manuscript.

**General Comments**

**The reviewer writes:** "ABC (both rejection and SMC) approximates likelihood in the following ways: Firstly, exact conformity to the the predictions of the model (theta) is relaxed using a distance measure and threshold. Secondly, if the simulation linking cause (theta) to observation (y) is stochastic then it uses a finite set of MC realisations instead of an integral over all outcomes of the random variables not of interest. Neither of these properties are used in the work presented here." **The reviewer confuses us with their statement: We relax exact conformity by using a distance measure and threshold (we use the Jaccard index as a distance measure between two topology graphs and allow an error threshold). The reviewer further states that** "Nothing what separates ABC from the traditional Bayesian approaches is used here.**", but then goes on to state that** "[…] it is ABC only in the most superficial sense**", providing us with a logical contradiction. It would have been helpful for us if the reviewer had provided us with literature references to properly understand their arguments.**

**We have mostly removed the use of the term "likelihood-free" when referring to ABC, although this is often stated as such in literature. We have also clarified our choice of prior parametrization, as we agree that this will help the reader better understand the experiments despite the focus of the paper on the method itself.**

**Specific Comments**

1) Remove claims all of circumventing or simplifying specification of topological knowledge due to ABC, these are untrue for the constraint presented here. I have highlighted these in the attached annotated pdf along with more detailed comments for each. Lines 60, 150, 155, 165.

**AR: We have reworded our description of topological knowledge to make it more clear to the reader how we acquired it (through interpretation). In the supplement the reviewer writes "**ABC is not a method for circumventing the subjectivity of specifying priors or likelihoods. It is likelihoodfree only in the sense of a likelihood never being explicitly calculated due to the nature of the approximation built into the samplers.", **which we agree with. We hope our changes clarify this to the reader, as we never intended to claim that using ABC over conventional Bayesian inference makes the method somehow more objective.**

**In the supplement the reviewer further notes that** "the implicit probability function you use is simply the one that assigns uniform probability to all theta within the region where d < iota.". **But as we *always* accept our parameter samples if they generate a model with a summary statistic within error, we would think that this does not constitute a uniform probability density function, as the area of the curve would be above 1.**

2) Remove all uses of the term 'likelihood-free'. There is no place in this article where its use helps clarify how the proposed approach works. Its only effect is as a potential source of misinterpretation. To avoid unneeded additional review rounds I am asking for complete removal and not fixing its use.

**AR: We have removed the term "likelihood-free" in the manuscript to avoid confusion of the reader. It remains in two places where we directly refer to literature that describes Bayesian optimization for likelihood-free inference (Gutmann and Corander, 2016) and where we point the reader to literature that specifically describes ABC as a "likelihood-free" method (Marin et al, 2012). Again, it would have been really helpful for us if the reviewer had provided literature references for us to understand their issue with this specific term.**

3) Do not refer to the initial connectivity graph as an observation. It is a semi-subjectice semi-empirically derived parameter to a subjectively chosen constraint family. Describe briefly how it is obtained and what the reasoning behind using the 'initial' graph was.

**AR: We have referred to the initial topology graph as an observation, as this is the terminology used in literature. We understand that this could potentially lead to certain confusion with the reader. We have thus added the clarification to the manuscript, that we treat our subjectively chosen topology graph as the observation in the terminology of ABC.**

4) I am not asking that you replace 'likelihood' with 'prior' when referring to your topological constraint. Instead, please state somewhere that you choose to go with this label but that it could also be considered a prior or empirical prior and that the application of these terms is not always clear cut. State that you are simply treating the adjacency graph (y) as an observation.

**AR: We already describe the topology graph (adjacency graph) as an observation --- something the reviewer has advised against in their previous comment.**

5) Two topology distance measures are defined in section 2.4 which are never used. Since they are never discussed, analysed or compared, they serve no purpose. I suggest you remove them to simplify and shorten the already long paper, but feel free to ignore this suggestion.

**AR: They are not used, but we think it serves the purpose of helping the reader think about different possible approaches to constrain geomodels using graph structures --- which we think has value.**

6) Remove the mention of fuzzy sets in section 2.5, it is not relevant. Your posterior is probabilistic not fuzzy. It represents degrees of certainty concerning a single underlying truth, not degrees of membership to a category.

**AR: We have removed the detailed explanations of information entropy in Section 2.5 and refer the reader to the relevant literature instead. Information entropy is used for assessing the prior and approximate posterior geomodel grids, not the posterior parameter distributions themselves.**

7) Several sentences need to be reworded for clarity or readability. I have highlighted these in the annotated pdf. Please try to address most of them.

**AR: We have reworded many parts of the manuscript given the suggestions by both reviewers.**

**References**

[revised manuscript text omitted]

---

## Author Response (AR2)

**Response to RC1 - Ashton Krajnovich**
**AR = Authors Response**

Thank you very much for the positive feedback and finding typos. We have corrected the text on the indicated locations.

**Response to RC2 - Anonymous Reviewer**
**AR = Authors Response**

*General Comments*:

**The reviewer writes**: "The manuscript still reads like a draft version"

**AR:** After careful evaluation, we found some small typos that were fixed. We do not see this comment justified. Surely, different opinions about writing styles exist and the reviewers can use their preferred style in their manuscript.
* * *
**The reviewer writes:** " #3 has been ignored." (original #3 comment:  Terms like observation, prior and likelihood are applied in an unclear manner often contradicting convention)

**AR:** After pondering about this comment for quite some time, we think that it relates to a misunderstanding about the application of probabilistic inference in this paper.

We do not see Bayes' equation as a strict separation between subjective knowledge and data, but as a way to combine conditional probabilities. This aspect is, in essence, obvious from the definition of conditional probabilities in the derivation of the Bayes equation. Of course, this is not our invention - but a mainstream view that is encapsulated in the use of probabilistic hierarchical models (e.g. Koller and Friedman, 2009). Also, we would like to refer to the interesting perspective in Nearing & Gupta (2018). We also included a clarification about this aspect in the manuscript (P6 L6).

We are still not sure if this is the point that led to the strong comments by the reviewer. It seems to be the case that this is a subject of philosophical debate for some - and that's surely fine and important, but outside of the topic of this paper and we firmly believe that any strong opinions in this direction should not hinder the publication about the aspect that *this* paper is about: including information about topological relationships in geological modeling frameworks. In the aim to stress the main theme of the paper, we have restructured the introduction with a stronger emphasis on how topological constraints can be used to encode geological knowledge in structural geology probabilistic inferences.

Koller, D., & Friedman, N. (2009). Probabilistic graphical models: principles and techniques. MIT press.

Nearing, G.S., Gupta, H.V. Ensembles vs. information theory: supporting science under uncertainty. Front. Earth Sci. 12, 653–660 (2018). https://doi.org/10.1007/s11707-018-0709-9
* * *
***Specific Comments***

**The reviewer writes: "**I could not find the place in the revised manuscript where this change was made". (original comment: Do not refer to the initial connectivity graph as an observation. It is a semi-subjectice semi-empirically derived parameter to a subjectively chosen constraint family. Describe briefly how it is obtained and what the reasoning behind using the 'initial' graph was.
)

**AR:** When we use the term observation we refer to the parameter *y* of the Bayes Eq. 2 not to the literal semantic meaning of the word observation. Since it may lead to confusion to some readers we have added some extra clarification when observation is defined:

*Notice that when the words "observation" or "observed data" are used in the context of a probabilistic model, we refer to this mathematical term $y$ instead to the literal semantic meaning of the words.*

In addition we add a clarification about why derived the term "y" from the initial geological model topology:

*The assumption is that this topological graph encapsulates some of the geological knowledge used during its construction by an expert and thus, geometrical configurations more similar to this graph can be considered more likely. This graph would be treated from this point on as a "observation" $y$ due to its use as a constraint within the probabilistic model.*
* * *
**The reviewer writes:** This comment has not been addressed. All I asked is that you state that the initial graph is what is treated as an observation even though its derivation includes significant subjective steps. This ties in with specific comment 3 as the procedure behind the observation was never properly discussed. (original comment: I am not asking that you replace 'likelihood' with 'prior' when referring to your topological constraint. Instead, please state somewhere that you choose to go with this label but that it could also be considered a prior or empirical prior and that the application of these terms is not always clear cut. State that you are simply treating the adjacency graph (y) as an observation.)

**AR:** We disagree that using the topological graph as the "observation" is a choice and could be considered as prior. In fact, it is in the nature of implicit representations that topology is not fixed (e.g. Wellmann and Caumon, 2018). The mathematical model M, used to relate model parameters, $p(\theta)$ and observations also limits which information (data) can be used as

model parameters or observation---analogous to an inverse problem. Topology---similar to some geophysical data---is a good example of information that cannot easily be used directly as input in this type of geological modeling algorithm.

There are many reasons to favour one probabilistic model over others. In this paper the main reason to select this specific probabilistic model is to obtain a set of parameters and parameter correlations capable of generating valid geological models---for the interpolation function used during the inference.

Someone could be tempted to think that if a model does not support information derived from human interpretation as a form of prior parameters, θ (as opposed to "observations", y), that this should be a clear indication that the selected model must be inadequate. However, in the authors opinion, such a distinction between "measured observations" and "human guesses" seems to treat the Bayesian equality as something more fundamental other than what it is: a mathematical tool to perform inference.

Gelman, A. and Hennig, C. (2017), Beyond subjective and objective in statistics. J. R. Stat. Soc. A, 180: 967-1033. https://doi.org/10.1111/rssa.12276
* * *
**The reviewer writes:** Still incorrect as I stated in the previous annotated document. (original: ABC does not learn prior distributions, it approximates posteriors over model parameters.)

**AR:** Reworded to be extra precise:

*We demonstrate how we can infer the posterior distributions of the model parameters using topology information in two experiments*
* * *
**The reviewer writes:** Likelihood relates data to model parameters. This reads if the likelihood function is that data. The function is a relation not data in itself.

**AR:** Agree, this paragraph can be misleading. We have reworded the sentences involved:

*In other words, by conditioning the probability of model parameters to some additional data, we are able to increase the overall information of the probabilistic model. Additional data can be, for example, a range of possible layer thicknesses in a depositional setting, geophysics or arguably geological knowledge in the form of valid geometrical configurations.*

*While the overall idea has been demonstrated in some specific cases, the general question of how to define suitable likelihood functions for specific type of observations---given a specific geological systems and diverse types of prior geological knowledge---still remains.*
* * *
**The reviewer writes:** Using ABC to indirectly specify a likelihood function is your choice. The claim that obtaining such a function is intractable is untrue for the one you use.

If 'theta' is the model parameters and 'y' is the initial graph. Define a binary statistic of 'theta', call it 'x', as true if the topology graph of 'theta' is within distance 'iota' of 'y', and false otherwise. If we then formulate the likelihood in terms of 'x' being the observation 'f(x|theta)' and sample from posterior 'p(theta|x)', this samples from the same posterior as what your ABC method does in algorithm 1 and approximates in algorithm 2. Replacing data with a statistic of it to simplify a problem is a very old practice. So I will repeat this, the claim that the use of ABC solves an intractable likelihood specification is not true for this specific work.

**AR:** It is true that the problem can be formulated using a likelihood function instead of an ABC distance function. Arguably most of the ABC Inferences could be constructed using likelihood functions for that matter. Probabilistic or not, many functions have the necessary properties to perform inference. In any case, since this is not the subject of the paper we have reformulated the sentence trying to demystify the use of ABC:

*The origin of topological information is generally qualitative. For this reason, choosing a likelihood function, trying to connote any probabilistic meaning to the comparison of topological graphs, does not seem to enhance the inference \citep{curtis_optimal_2004}. This work, favouring model simplicity, adopts an Approximate Bayesian Computation (ABC) approach to compute the posterior using a distance function instead of a likelihood function.*
* * *
**The reviewer writes:** The definition of model parameters is still too vague and poorly written. The author conflates parameters with probability distributions as I pointed out in the previous annotated document.

**AR:** We refer here to parameters of the interpolation function, not the parameters of the probability distributions. Any geomodeling interpolator requires different types of parameters---geometric parameters (e.g. x coordinate), abstract parameters (e.g. the degree of a Matérn kernel used for one of several scalar fields used for the interpolation), and semantic parameters (e.g. faults). We refer the reviewer to Wellmann & Caumon, 2018, for a more detailed overview. Due to this level of complexity, we need to select a subset of the geomodeling parameters to be stochastic. We do not claim to perform an exhaustive analysis of all possible uncertainty in this manuscript.

We clarified this aspect in the text to avoid further confusion:
For case study 1, Figure 4 b) and Table 1 shows which geological modeling parameters are function of the probabilistic model parameters -- i.e. prior distributions of the Gaussian Family defined by the parameters mean and standard deviation.

For case study 2, since the geometry is harder (thus the existence of case 1) we only provide Table 2. However, the nature of the probabilistic model parameters are exactly the same as case study 1: *(i) vertical location of the layer interfaces for within each fault block; (ii) the lateral location of the fault interfaces.* (P12 L10 and P11 L3)
* * *
**The reviewer writes:** "chosen emperically" how?

**AR:** Added some extra information about the threshold value:

*The initial topology graph is used as a constraining summary statistics using*
*ABC with rejection sampling (ABC-REJ) using a threshold of $\epsilon=0.025$.*
*The absolute threshold value will be directly proportional to the sensitivity of the model*
*geometry with respect to the stochastic parameters. This prevents the selection of a value*
*independent of the actual geological model under study. In this case study, the value of*
*$\epsilon$ has been chosen empirically by performing several predictive simulations.*
*Results were evaluated based on their correspondence to the geological setting.*
* * *
**The reviewer writes:** "while reducing the number of required iterations through use of advanced sampling techniques."

Sampling technique efficiency was not presented as the focus in the preceding parts of this paper and should not as there is very little here on that topic

**AR:** Removed the sentence about sampling techniques.